# BREAKING THE TOTAL VARIANCE BARRIER: SHARP SAMPLE COMPLEXITY FOR LINEAR HETEROSCEDASTIC BANDITS WITH FIXED ACTION SET

Heyang Zhao[1][*]  Tianyuan Jin[2][*]  Weixin Wang[3]  Vincent Y. F. Tan[2]  Pan Xu[3,4,5]  Quanquan Gu[1]

[1]Department of Computer Science, University of California, Los Angeles
[2]Department of Electrical and Computer Engineering, National University of Singapore
[3]Department of Biostatistics & Bioinformatics, Duke University
[4]Department of Computer Science, Duke University
[5]Department of Electrical and Computer Engineering, Duke University
{hyzhao,qgu}@cs.ucla.edu
{tianyuan, vtan}@nus.edu.sg
{weixin.wang, pan.xu}@duke.edu

## ABSTRACT

Recent years have witnessed increasing interests in tackling heteroscedastic noise in bandits and reinforcement learning (e.g., Zhou et al., 2021; Zhao et al., 2023a; Jia et al., 2024; Pacchiano, 2025). In these works, the cumulative variance of the noise $\Lambda = \sum_{t=1}^{T} \sigma_t^2$, where $\sigma_t^2$ is the variance of the noise at round $t$, is used to characterize the statistical complexity of the problem, yielding *simple regret* bounds of order $\tilde{\mathcal{O}}(d\sqrt{\Lambda/T^2})$ for $d$-dimensional linear bandits with heteroscedastic noise (Zhou et al., 2021; Zhao et al., 2023a). However, with a closer look, $\Lambda$ remains the same order even if the noise is close to zero at half of the rounds, which indicates that the $\Lambda$-dependence is not optimal.

In this paper, we revisit the stochastic linear bandit problem with heteroscedastic noise, where the action set is prefixed throughout the learning process. We propose a novel variance-adaptive algorithm VAEE (Variance-Aware Exploration with Elimination) for large action set, which actively explores actions that maximizes the information gain among a candidate set of actions that are not eliminated. With the active-exploration strategy, we show that VAEE achieves a *simple regret* with a nearly *harmonic-mean* dependent rate, i.e., $\tilde{\mathcal{O}}\left(d\left[\sum_{t=1}^{T} \frac{1}{\sigma_t^2} - \sum_{i=1}^{\tilde{O}(d)} \frac{1}{[\sigma^{(i)}]^2}\right]^{-\frac{1}{2}}\right)$[1] where $\sigma^{(i)}$ is the $i$-th smallest variance among $\{\sigma_t\}_{t=1}^{T}$. For finitely many actions, we propose a variance-aware variant of G-optimal design based exploration, which achieves a simple regret of $\tilde{\mathcal{O}}\left(\sqrt{d \log |\mathcal{A}|}\left[\sum_{t=1}^{T} \frac{1}{\sigma_t^2} - \sum_{i=1}^{\tilde{O}(d)} \frac{1}{[\sigma^{(i)}]^2}\right]^{-\frac{1}{2}}\right)$. We also establish a nearly matching lower bound for the fixed action set setting indicating that *harmonic-mean* dependent rate is unavoidable. To the best of our knowledge, this is the first work that breaks the $\sqrt{\Lambda}$ barrier for stochastic linear bandits with heteroscedastic noise.

## 1 INTRODUCTION

The stochastic multi-armed bandit (MAB) problem is a fundamental framework for studying the exploration-exploitation trade-off in sequential decision-making (Auer et al., 2002). In the classic stochastic bandit setting, an agent repeatedly selects an arm from a set of arms and receives a

---

[*]Equal contribution.

[1]The formal notation is given by $\tilde{\mathcal{O}}\left(d\left[\sum_{t=1}^{T} \frac{1}{\sigma_t^2} - \sum_{i=1}^{\iota(d,T)} \frac{1}{[\sigma^{(i)}]^2}\right]^{-\frac{1}{2}}\right)$, where $\iota(d,T) = \tilde{O}(d)$ is a function of $d$ and $T$. For simplicity, we use $\tilde{O}(d)$ to denote $\iota(d,T)$ throughout the paper.

stochastic reward associated with the chosen arm. The goal of the agent is to maximize the cumulative reward over a series of rounds by balancing exploration (trying out different arms to gather information) and exploitation (choosing the best-known arm based on past observations). Over the past few decades, various algorithms have been proposed to tackle the stochastic bandit problem from the perspectives of minimax optimal sample complexity (Audibert & Bubeck, 2009; Ménard & Garivier, 2017; Jin et al., 2021; 2023).

To further leverage the heteroscedastic nature of the noise in real-world applications, recent works have extended the classic bandit framework to account for heteroscedastic noise, where the variance of the noise can vary across different arms and time steps (Zhou et al., 2021; Zhao et al., 2023a; Jia et al., 2024; Pacchiano, 2025). These works have shown that by taking into account the varying variance of the noise, it is possible to design more efficient algorithms that achieve better performance in terms of regret bounds. In detail, Zhou et al. (2021) first considered the linear bandit problem with heteroscedastic noise and proposed a variance-aware algorithm that achieved a regret bound of order $\tilde{\mathcal{O}}(d\sqrt{\Lambda} + \sqrt{dT})$, where $\Lambda = \sum_{t=1}^{T} \sigma_t^2$ is the cumulative variance of the noise along $T$ time steps and $d$ is the dimension of the feature space. Later, Zhou & Gu (2022) improved the cumulative regret bound to $\tilde{\mathcal{O}}(d\sqrt{\Lambda} + d)$, yielding a simple regret[2] bound of order $\tilde{\mathcal{O}}(d\sqrt{\Lambda/T^2} + d/T)$. More recently, Jia et al. (2024) proposed `VarCB`, which achieves a tighter cumulative regret bound of order $\tilde{O}(\sqrt{|\mathcal{A}|\Lambda d} + d^2)$ for contextual bandits with a fixed action set, where $|\mathcal{A}|$ is the size of the action set and $\Lambda = \sum_{t=1}^{T} \sigma_t^2$ is the variance budget, and further extended their analysis to general function classes. In the same work, they proved a minimax lower bound of order $\tilde{\Omega}(\sqrt{\min(|\mathcal{A}|, d) \cdot \Lambda} + d)$ when $d \le \sqrt{|\mathcal{A}|T}$, showing that the $\sqrt{\Lambda}$ dependence is unavoidable in the worst case over instances and variance sequences. Recently, He & Gu (2025) further established (up to logarithmic factors) matching variance-dependent lower bounds of order $\tilde{\Omega}(d\sqrt{\Lambda})$ for linear contextual bandits with time-varying action sets and arbitrary variance sequences, confirming that the $\sqrt{\Lambda}$ scaling is information-theoretically optimal even when the entire variance sequence is revealed to the learner. On the other hand, He & Gu (2025) proved that for stochastic linear bandits where the action set is prefixed, the $\tilde{\Omega}(d\sqrt{\Lambda})$ lower bound does not hold. This motivates us to pursue sharper variance-dependent regret bounds for stochastic linear bandits with a fixed action set (either finite or infinite).

More specifically, existing regret bounds depend on the total variance term $\Lambda$, which overlooks the heterogeneity of information gain across actions and time steps with different noise levels. Consider an extreme case: if $\sigma_t^2 \approx 0$ for all $t \le t_0$ with $t_0 = \tilde{O}(d) \ll T$, the $d$-dimensional weight parameter in the linear bandit problem could be recovered almost exactly. In such a case, the regret bound should be essentially independent of the noise variance after time step $t_0$. This motivates an important open question in heteroscedastic stochastic linear bandits:

*Can we improve upon the $\sqrt{\Lambda}$ dependence in the regret bounds in stochastic linear heteroscedastic bandits?*

In this paper, we revisit the problem of best-arm identification in stochastic linear bandits under heteroscedastic noise, where the action set is prefixed and the variances of the reward distribution may vary significantly across actions. The primary performance metric we focus on is the simple regret, which measures the suboptimality of the action recommended after a fixed budget of exploration. Our results highlight the fundamental role of the harmonic mean of the variances in characterizing the attainable regret rate.

Our main contributions are summarized as follows:

- **Variance-adaptive exploration for large action sets.** We propose a novel algorithm, `VAEE` (Variance-Aware Exploration with Elimination), designed to handle large (potentially infinite) action sets. The key idea is to maintain a candidate set of promising actions and actively explore those that maximize the information gain subject to elimination rules. We prove that `VAEE`

---

[2]Simple regret quantifies the expected gap between the optimal reward and the reward of the arm proposed by the algorithm.

Table 1: Comparison between different algorithms for stochastic (linear) contextual bandits. Here $d$ is the feature dimension, $T$ is the number of rounds, $\{\sigma_t\}_{t \in [T]}$ is the variance of noise at round $t \in [T]$, $|\mathcal{A}|$ is the size of the arm set, and $\Lambda = \sum_{t=1}^T \sigma_t^2$ is the cumulative variance of the noise. The time-varying means that the action set is allowed to change over time (possibly chosen in advance by an oblivious adversary), whereas fixed means that the same action set is used in all rounds. The infinite means the action set may contain infinitely many actions, whereas finite means the action set contains only finitely many actions. The lower bounds from Jia et al. (2024); He & Gu (2025) are derived for the cumulative regret. We convert them to be comparable to our simple regret by dividing them by $T$. Note that their lower bounds are derived for the worst case sequence of noise variance, while our lower bound has a refined dependence on the noise variance sequence.

| Algorithm | Simple Regret Upper Bound | Simple Regret Lower Bound | Action Set |
|---|---|---|---|
| Weighted OFUL (Zhou et al., 2021) | $d\sqrt{\Lambda/T^2}$ | - | Time-varying/Infinite |
| Weighted OFUL+ (Zhou & Gu, 2022) | $d\sqrt{\Lambda/T^2}$ | - | Time-varying/Infinite |
| VOFUL (Zhang et al., 2021) | $d^{9/2}\sqrt{\Lambda/T^2}$ | - | Time-varying/Infinite |
| VOFUL2 (Kim et al., 2021) | $d^{3/2}\sqrt{\Lambda/T^2}$ | - | Time-varying/Infinite |
| SAVE (Zhao et al., 2023a) | $d\sqrt{\Lambda/T^2}$ | - | time-varying/Infinite |
| LinNATS (Xu et al., 2023) | $d^{3/2}\sqrt{\Lambda/T^2}$ | - | Time-varying/Infinite |
| VarCB (Jia et al., 2024) | $\sqrt{|\mathcal{A}|\Lambda d/T^2}$ | $\Omega\big(\sqrt{\min(|\mathcal{A}|,d)\Lambda/T^2}\big)$ | Time-varying/Finite |
| He & Gu (2025) | - | $\tilde{\Omega}\big(d\sqrt{\Lambda/T^2}\big)$ | Time-varying/Infinite |
| **VAEE (Ours)** | $d\Big[\sum_{t=1}^T \frac{1}{\sigma_t^2} - \sum_{i=1}^{\tilde{O}(d)} \frac{1}{[\sigma^{(i)}]^2}\Big]^{-\frac{1}{2}}$ | $\Omega\Big(d\big(\sum_{i=1}^t \frac{1}{\sigma_i^2}\big)^{-\frac{1}{2}}\Big)$ | Fixed/Infinite |
| **VAGD (Ours)** | $\sqrt{d\log|\mathcal{A}|}\Big[\sum_{t=1}^T \frac{1}{\sigma_t^2} - \sum_{i=1}^{\tilde{O}(d)} \frac{1}{[\sigma^{(i)}]^2}\Big]^{-\frac{1}{2}}$ | - | Fixed/Finite |

achieves a simple regret bound of

$$\tilde{\mathcal{O}}\left( d\left[ \sum_{t=1}^T \frac{1}{\sigma_t^2} - \sum_{i=1}^{\tilde{O}(d)} \frac{1}{[\sigma^{(i)}]^2} \right]^{-\frac{1}{2}} \right),$$

where $d$ is the feature dimension, and $\{[\sigma^{(i)}]^2\}$ are the ordered list of the variance sequence $\{\sigma_t^2\}$. This establishes a nearly harmonic-mean dependent rate for the simple regret.

- **Variance-aware G-optimal design for finite action sets.** For the case of a finite action set $\mathcal{A}$, we propose a variance-adaptive variant of G-optimal design based exploration. We show that this strategy achieves a simple regret bound with improved dependence on the dimension $d$ as follows

$$\tilde{\mathcal{O}}\left( \sqrt{d\log|\mathcal{A}|} \left[ \sum_{t=1}^T \frac{1}{\sigma_t^2} - \sum_{i=1}^{\tilde{O}(d)} \frac{1}{[\sigma^{(i)}]^2} \right]^{-\frac{1}{2}} \right).$$

- **Lower bound matching the harmonic-mean rate.** We establish a nearly matching lower bound for the fixed-action setting, showing that the harmonic-mean dependence is intrinsic to the problem. This demonstrates that our algorithms are essentially optimal in their variance dependence.

- **Breaking the $\sqrt{\Lambda}$ barrier.** To the best of our knowledge, this is the first work that surpasses the classical $\sqrt{\Lambda}$-type dependence in simple regret bounds for linear bandits with heteroscedastic noise, where $\Lambda$ denotes the variance proxy commonly used in prior analyses. A comprehensive comparison on the simple regret bounds is provided in Table 1 for the reader's reference.

**Notations.** We use bold lowercase letters (e.g., $\mathbf{a}$) to denote vectors and bold uppercase letters (e.g., $\mathbf{A}$) to denote matrices. For a vector $\mathbf{a} \in \mathbb{R}^d$, we use $\|\mathbf{a}\|_2$ to denote its Euclidean norm. For a positive definite matrix $\mathbf{A} \in \mathbb{R}^{d \times d}$, we define the elliptical norm of a vector $\mathbf{a}$ as $\|\mathbf{a}\|_{\mathbf{A}} = \sqrt{\mathbf{a}^\top \mathbf{A}\mathbf{a}}$. We use $\mathbf{I}_d$ to denote the $d \times d$ identity matrix. For a set $\mathcal{A}$, we use $|\mathcal{A}|$ to denote its cardinality. We use $\tilde{O}(\cdot)$ to hide logarithmic factors in $d, T, 1/\delta, 1/\sigma_{\min}, 1/\sigma_{\max}$. For a sequence $\{a_t\}_{t=1}^T$, we use $a^{(i)}$ to denote the $i$-th smallest element in the sequence.

## 2 RELATED WORK

**Variance-Aware Regret for Linear Bandits with Heteroscedastic Noise.** The incorporation of variance information in linear bandit algorithms has garnered significant attention in recent years,

leading to substantial improvements in regret bounds. Early work by Kirschner & Krause (2018) introduced the concept of information-directed sampling for bandits with heteroscedastic noise, demonstrating that leveraging variance information can lead to more efficient exploration strategies. Later, Zhou et al. (2021) proposed a variance-aware algorithm for linear bandits that achieves a regret bound of order $\tilde{\mathcal{O}}(d\sqrt{\Lambda} + \sqrt{dT})$, where $\Lambda = \sum_{t=1}^{T} \sigma_t^2$ is the cumulative variance of the noise. This result was further improved by Zhou & Gu (2022) to a tighter bound of $\tilde{\mathcal{O}}(d\sqrt{\Lambda} + d)$. Zhao et al. (2023b) later proposed a peeling-based algorithm that achieves a similar regret bound.

The challenge of unknown conditional variances has been addressed by several researchers. Zhang et al. (2021) and Kim et al. (2021) developed algorithms that operates without prior knowledge of the variance, achieving regret bounds that adapt to the observed noise levels. However, these approaches are not tractable for large action sets and incur sub-optimal dependence on $d$. Zhao et al. (2023a) proposed a computationally efficient algorithm that achieves a regret bound of order $\tilde{\mathcal{O}}(d\sqrt{\Lambda} + d)$ without requiring prior knowledge of the variances.

More recently, Pacchiano (2025) extended the variance-aware framework of Zhao et al. (2023a) to the general function approximation setting, achieving a regret bound of order $\tilde{\mathcal{O}}(d_{eluder}\sqrt{\log(\mathcal{F})\Lambda} + d_{eluder}\log(\mathcal{F}))$, where $d_{eluder}$ is the eluder dimension and $\log(\mathcal{F})$ is the log-covering number of the function class $\mathcal{F}$. Concurrently, Jia et al. (2024) introduced `VarCB`, an algorithm that attains a regret bound of $\tilde{\mathcal{O}}(\sqrt{|\mathcal{A}|\Lambda d} + d^2)$ for bandits with few actions, and extended their results to general function classes. They also established a worst-case lower bound of order $\Omega(\sqrt{\min(|\mathcal{A}|, d)\Lambda} + d)$ when $d \leq \sqrt{|\mathcal{A}|T}$. He & Gu (2025) further studied the setting where the action set can change arbitrarily over time and proved an instance-dependent lower bound of order $\Omega(d\sqrt{\Lambda}/\log T)$ for the expected cumulative regret. These results indicate that the $\sqrt{\Lambda}$ dependence is unavoidable in such settings.

**Bandits with Heavy-Tailed Noise.** The topic of robustness to heavy-tailed rewards has received considerable attention in recent years, addressing the limitations of classical bandit algorithms that assume sub-Gaussian or bounded noise. Bubeck et al. (2013) pioneered this research direction by studying heavy-tailed rewards in multi-armed bandits, establishing that standard concentration inequalities fail in such environments. For linear bandits, Medina & Yang (2016) proposed truncation-based methods and median-of-means estimators to handle heavy-tailed noise, achieving sublinear regret bounds. Shao et al. (2018) adopted median-of-means techniques with a well-designed allocation of decisions to achieve nearly optimal regret bounds. Later, Xue et al. (2020) introduced a SupLin-based algorithm (Chu et al., 2011) which further improved the dimension dependence in the regret bounds. More related works include Li & Sun (2024), Huang et al. (2023), which proposed Huber regression based algorithms to handle heteroscedastic heavy-tailed noise. Recently, Ye et al. (2025) proposed a Catoni's estimator based algorithm that achieves adaptive regret bounds in bandits with general function approximation.

**Variance-Dependent Bounds in MDPs.** As a natural extension of bandits, Markov Decision Processes (MDPs) have also been studied under the lens of variance-dependent regret bounds. In tabular MDPs, Zanette & Brunskill (2019) first established a variance-dependent regret bound which scales with the square root of the maximum variance of the value function. Afterwards, Zhou et al. (2023) proposed MVP-V, an algorithm that achieves a regret bound scaling with the square root of the total variance of the value function, achieving worst-case optimal regret bound. In MDPs with linear function approximation, Zhao et al. (2023a) proposed a variance-aware algorithm that achieves a second-order and horizon-free regret bound. More recently, there have been several works (Wang et al., 2024; Zhao et al., 2024; Wang et al., 2025; Zhao et al., 2025) presenting variance-dependent regret bounds in MDPs with general function approximation.

## 3 PRELIMINARIES

We consider a heteroscedastic variant of the stochastic linear bandit problem. Let $T$ be the total number of rounds. The action set $\mathcal{A}$ is fixed. At each round $t \in [T]$, the interaction between the agent and the environment is as follows:

1. The agent selects $\mathbf{a}_t \in \mathcal{A}$ based on the past observations $\mathcal{F}_{t-1} = (\boldsymbol{a}_1, r_1, \ldots, \boldsymbol{a}_{t-1}, r_{t-1})$ up to time $t-1$.
2. The environment generates the stochastic noise $\eta_t$ at round $t$ and reveals the stochastic reward $r_t = \langle \boldsymbol{\theta}^*, \mathbf{a}_t \rangle + \eta_t$ to the agent.
3. The agent observes the variance of $\eta_t$.

We assume that for all $\mathbf{a} \in \mathcal{A}$, it holds that $\|\mathbf{a}\|_2 \leq 1$ and $\|\boldsymbol{\theta}^*\|_2 \leq 1$.

**Remark 3.1.** Our assumption that the action set is fixed is necessary for achieving the harmonic-mean dependent rate. In scenarios where the action set can change arbitrarily over time, it is possible to construct instances where the cumulative variance $\Lambda = \sum_{t=1}^{T} \sigma_t^2$ remains the most appropriate measure of statistical complexity. This is because an adversarially chosen action set can force the algorithm to repeatedly explore less informative actions when the noise level is low, thereby negating the benefits of a harmonic-mean based approach. A detailed study of this phenomenon is provided by He & Gu (2025), which demonstrates that in the case of adversarially changing contexts, there exists a lower bound of order $\Omega(d\sqrt{\Lambda}/\log T)$ for the expected cumulative regret, indicating that the $\sqrt{\Lambda}$ dependence is unavoidable in such settings.

Therefore, to fully leverage the advantages of our proposed variance-adaptive algorithms and achieve the improved regret bounds, we focus on the standard stochastic linear bandit setting (Lattimore & Szepesvári, 2020) where the action set is fixed throughout the learning process.

We introduce the following assumption on the noise $\eta_t$.

**Assumption 3.2.** The noise $\eta_t$ is conditionally $\sigma_t$-sub-Gaussian, i.e., for all $\lambda \in \mathbb{R}$, it holds that $\mathbb{E}\left[\exp\left(\lambda\eta_t\right) \mid \mathcal{F}_{t-1}\right] \leq \exp(\lambda^2\sigma_t^2/2)$, where $\mathcal{F}_{t-1}$ is the filtration up to round $t-1$. We assume that there exist known constants $\sigma_{\min}, \sigma_{\max} > 0$ such that $\sigma_{\min} \leq \sigma_t \leq \sigma_{\max}$ for all $t \in [T]$.

**Remark 3.3.** This assumption follows from the original formulation of heteroscedastic bandits by Kirschner & Krause (2018). Later works (Zhou et al., 2021; Zhao et al., 2023a; Jia et al., 2024) have slightly generalized this assumption to only require the variance of $\eta_t$ to be bounded by $\sigma_t^2$ and the magnitude of $\eta_t$ to be bounded by a constant. However, this generalization does not significantly affect our analysis or results, as we will discuss in Appendix E that our algorithms can be extended to handle heavy-tailed noise by replacing the least-squares estimator with a robust estimator.

In this paper, we focus on the best-arm identification problem in linear bandits with heteroscedastic noise. The performance of an algorithm is measured by the simple regret defined as follows:

$$\mathrm{SR}(T) = \mathbb{E}\left[\max_{\mathbf{a}\in\mathcal{A}}\langle\boldsymbol{\theta}^*, \mathbf{a}\rangle - \langle\boldsymbol{\theta}^*, \hat{\mathbf{a}}_T\rangle\right], \tag{3.1}$$

where $\hat{\mathbf{a}}_T$ is the action recommended by the algorithm after $T$ rounds of exploration.

**Remark 3.4.** In the stochastic linear bandit literature, simple regret is closely connected to cumulative regret. In particular, if an algorithm achieves cumulative regret of order $\tilde{\mathcal{O}}(\sqrt{dT})$, then its simple regret can be shown to be of order $\tilde{\mathcal{O}}(\sqrt{d/T})$ (Lattimore & Szepesvári, 2020). In the heteroscedastic setting, however, the varying and unpredictable noise levels make this relationship more subtle. For example, a harmonic-mean dependence of the simple regret on the variances does not necessarily translate to the same dependence for cumulative regret. In this work, we therefore focus on directly analyzing the simple regret of our proposed algorithms.

# 4 STOCHASTIC LINEAR BANDITS WITH INFINITE ACTION SPACE

In this section, we propose Variance-Aware Exploration with Elimination (`VAEE`), a variance-adaptive approach designed for linear bandits operating in environments with heteroscedastic noise and potentially large action spaces. The algorithm is displayed in Algorithm 1, which builds upon the Optimism in the Face of Uncertainty for Linear bandits (OFUL) framework while incorporating variance information to improve exploration efficiency and regret bounds.

**Variance Adaptation.** The algorithm explicitly incorporates variance information $\sigma_t$ observed at each time step and uses variance-weighted updates for both the covariance matrix and parameter estimation (Zhou et al., 2021). This allows the algorithm to adaptively adjust its confidence sets based on the observed noise levels, leading to more accurate estimates of the underlying parameters.

**Active Exploration.** Algorithm 1 employs an active exploration strategy that selects actions based on their potential to maximize information gain. Specifically, at each round, the algorithm chooses the action that maximizes the uncertainty in the parameter estimate, as measured by the Mahalanobis distance with respect to the inverse covariance matrix. This encourages exploration of actions that are expected to provide the most informative feedback.

---

**Algorithm 1** Variance-Aware Exploration with Elimination (`VAEE`)

---

**Require:** $\mathcal{A} \subset \mathbb{R}^d, \delta$.
1: Initialize $V_0 \leftarrow \lambda I_d, \hat{\boldsymbol{\theta}}_0 \leftarrow 0, \mathcal{A}_1 \leftarrow \mathcal{A}$.
2: **for** $t = 1, \ldots, T$ **do**
3:     Pull the action $\mathbf{a}_t \leftarrow \max_{\mathbf{e} \in \mathcal{A}_t} \|\mathbf{e}\|_{V_{t-1}^{-1}}$.
4:     The agent receives the reward $r_t$ and the variance $\sigma_t$.
5:     Calculate $V_t \leftarrow V_{t-1} + \sigma_t^{-2} \mathbf{a}_t \mathbf{a}_t^\top$.
6:     Calculate $\hat{\boldsymbol{\theta}}_t \leftarrow V_t^{-1} \sum_{s=1}^t \sigma_s^{-2} \mathbf{a}_s r_s$.
7:     Set confidence set as follows $\mathcal{C}_t \leftarrow \{\boldsymbol{\theta} \mid \|\boldsymbol{\theta} - \hat{\boldsymbol{\theta}}_t\|_{V_t^{-1}}^2 \le \beta_t\}$.
8:     Eliminate low rewarding arms: $\mathcal{A}_{t+1} \leftarrow \{\mathbf{a} \in \mathcal{A}_t : \max_{\mathbf{e} \in A_t} \min_{\boldsymbol{\theta} \in \mathcal{C}_t} \langle \boldsymbol{\theta}, \mathbf{e} \rangle \le \max_{\boldsymbol{\theta} \in \mathcal{C}_t} \langle \boldsymbol{\theta}, \mathbf{a} \rangle \}$.
9: **end for**

---

### 4.1 CASE STUDY ON WHY WEIGHTED OFUL FAILS

We now present a two-dimensional case study to illustrate that *Weighted OFUL* (Zhou et al., 2021) fails for structural reasons rather than due to a loose analysis, especially when the variance sequence contains low-variance windows and the information for learning the $d$-dimensional parameter vector grows anisotropically across coordinates. In contrast, our variance-sequence-aware design reallocates exploration toward weak coordinates and achieves an instance- and variance-sequence-dependent bound. In the example below, our algorithm attains simple regret of order $\varepsilon \exp\left(-\Theta(\log T/\varepsilon^2)\right)$, whereas Weighted OFUL yields only $\varepsilon \exp\left(-\Theta(\log T)\right)$. When $\varepsilon = T^{-1/4}$, the simple regret of VAEE is sharper than that of Weighted OFUL by a factor of $T^{\Theta(\sqrt{T})}$.

**Setup.** Consider a two-dimensional linear bandit with action set $\mathcal{A} = \{e_1, e_2, x\}$, where $e_1 = (1, 0)$, $e_2 = (0, 1)$, and $x = (1 - \varepsilon, \varepsilon)$, and with true parameter vector $\boldsymbol{\theta}^* = e_1$. We take confidence radii satisfying $\beta_t \le \beta = \Theta(\sqrt{\log T})$ and set $\varepsilon = T^{-1/4}$. The variance profile contains a global window $W$ of length $L$ in which the noise variance is $\sigma_t^2 = T^{-\alpha}$ for all $t \in W$ with $\alpha \in (0, 1)$, while outside $W$ the variance is constant.

**Simplifying assumption.** To isolate the behavior along the second coordinate, we make the following simplifying assumption. We assume that *before* entering $W$, both algorithms have already collected enough information in the $e_1$ direction so that the estimation error in the first coordinate is negligible. This is justified because pulls of $e_1$ and $x$ both provide substantial information about the first coordinate, and both our method and Weighted OFUL sample $x$ (or $e_1$) frequently. As a result, information in the first coordinate dominates that in the second, and the dominant source of error comes from limited information along the second coordinate, which is therefore our focus.

**Weighted OFUL in the Low Variance Window.** In the low-variance window $W$ where $\sigma_t^2 = T^{-\alpha}$, each pull of an arm $a$ contributes $T^\alpha \langle a, e_2 \rangle^2$ units of information to the second coordinate.

*Case 1 (start with $e_2$).* If Weighted OFUL initially pulls $e_2$ in $W$, then each pull adds $T^\alpha$ units of second-coordinate information. After about $\log T/\varepsilon^2$ such units have been gathered, the second-coordinate error is at most $\varepsilon$. Since $\mu_x = \langle x, \boldsymbol{\theta}^* \rangle = 1 - \varepsilon$ and $\boldsymbol{\theta}^* \in \mathcal{C}_t$ w.h.p., we have $\mu_x \le \text{UCB}_x(t)$ and hence $\text{UCB}_x(t) \ge 1 - \varepsilon = 1 - o(1)$.

Moreover, after $m = \Theta(T^{-\alpha} \log T/\varepsilon^2)$ pulls of $e_2$ we have $mT^\alpha = \Theta(\log T/\varepsilon^2)$, so $\|e_2\|_{V_t^{-1}} = \Theta(\varepsilon/\sqrt{\log T})$ and therefore $\text{UCB}_2(t) = \langle e_2, \hat{\boldsymbol{\theta}}_t \rangle + \beta \|e_2\|_{V_t^{-1}} \le \varepsilon + O(\beta\varepsilon/\sqrt{\log T}) = O(\epsilon) = o(1)$ using $\varepsilon = T^{-1/4}$ and $\beta = \Theta(\sqrt{\log T})$. Thus $\text{UCB}_2(t) < \text{UCB}_x(t)$ and Weighted OFUL switches to selecting $x$.

*Case 2 (keep pulling $x$).* If instead Weighted OFUL keeps pulling $x$ throughout $W$, then each pull of $x$ contributes only $\varepsilon^2 T^\alpha$ to the $e_2$ direction, so after $L$ pulls the total second-coordinate information is at most $L\varepsilon^2 T^\alpha$.

**Simple Regret of Weighted OFUL.** Choose $L = c_L T^{-\alpha} \frac{\log T}{\varepsilon^4} \Rightarrow L\varepsilon^2 T^\alpha = c_L \log T$. By a standard Chernoff/Hoeffding concentration, the failure probability of recommending $\hat{a}_T = x$ decays as $\exp\big(-\Theta(\log T)\big)$, hence the simple regret satisfies $\mathrm{SR}(T) \leq \varepsilon \cdot \exp\big(-\Theta(\log T)\big)$.

**Simple Regret of Algorithm 1.** Since our algorithm pulls the arm with the largest exploration bonus (Line 3 of Algorithm 1), it allocates the window $W$ to $e_2$ and gains $LT^\alpha \asymp c_L \frac{\log T}{\varepsilon^4}$ units of second-coordinate information within $W$. By a standard concentration inequality, the failure probability of recommending $\hat{a}_T = x$ decays as $\exp\big(-\Theta(\log T/\varepsilon^2)\big)$, hence the simple regret satisfies $\mathrm{SR}(T) \leq \varepsilon \cdot \exp\big(-\Theta(\log T/\varepsilon^2)\big)$. Since $\varepsilon = T^{-1/4}$, our simple regret is significantly lower than that of Weighted OFUL by a factor of $T^{\Theta(\sqrt{T})}$.

## 4.2 THEORETICAL RESULTS FOR VAEE

We now present the main theoretical results for VAEE. The following theorem establishes a simple regret bound with harmonic-mean dependence.

**Theorem 4.1** (Simple Regret of VAEE). Set $\beta_t = 2\sqrt{\lambda} + 16\sqrt{\log(4t^2/\delta) \cdot d \log \frac{d\lambda + t\sigma_{\min}^{-2}}{d\lambda}}$ and $\lambda = 1$ in Algorithm 1. Let $\sigma_T^{(i)}$ be the $i$-th smallest element in $\{\sigma_\tau^2\}_{\tau=1}^T$. With probability at least $1 - \delta$, the simple regret of Algorithm 1 satisfies

$$\mathrm{SR}(T) = \tilde{O}(\sqrt{d}) \min_{1 \leq k \leq T+1} \left\{ x = \sqrt{\frac{\iota(T) - k + 1}{\sum_{i=k}^T \frac{1}{[\sigma_T^{(i)}]^2}}} \,\middle|\, x \in [\sigma_T^{(k-1)}, \sigma_T^{(k)}], k \in [T+1] \right\},$$

where $\iota(T) = 2d \log(1 + \sum_{\tau \in [T]} \sigma_\tau^{-2}/d)$.

**Remark 4.2.** Theorem 4.1 provides a simple regret bound for VAEE that depends on the harmonic mean of the variances $\sigma_t^2$. To see this, we can simplify the bound by substituting $k = \tilde{O}(d)$ and $\iota(T) = \tilde{O}(d)$, which yields the following simplified expression:

$$\mathrm{SR}(T) = \tilde{O}\left( d \left[ \sum_{t=1}^T \frac{1}{\sigma_t^2} - \sum_{i=1}^{\tilde{O}(d)} \frac{1}{[\sigma^{(i)}]^2} \right]^{-\frac{1}{2}} \right). \tag{4.1}$$

This bound in (4.1) highlights that the simple regret decreases as the harmonic mean of the variances increases, effectively capturing the influence of low-variance actions on the overall performance. Notably, this result breaks the traditional $\sqrt{\Lambda}$ barrier, where $\Lambda = \sum_{t=1}^T \sigma_t^2$, demonstrating that our variance-adaptive approach can achieve significantly better performance in environments with heteroscedastic noise.

**Remark 4.3.** It is worth noting that our harmonic-mean dependence is subtracted by the contribution of the $\tilde{O}(d)$ smallest variances. This subtraction is unavoidable due to the inherent difficulty of estimating a $d$-dimensional parameter, which requires at least $d$ well-explored actions. In the worst-case scenario, these $d$ actions may correspond to the smallest variances, and then it is impossible to achieve a near zero simple regret with only $T = O(1)$ rounds of noise-free exploration. Therefore, the subtraction term in our bound is necessary to account for this fundamental limitation.

Nonetheless, we can still show that our simple regret bound is strictly sharper than the $\sqrt{\Lambda}$-type bounds in prior works (Zhou et al., 2021; Zhao et al., 2023a; Jia et al., 2024). To see this, we observe that $\min_{1 \leq k \leq T+1} \left\{ x = \sqrt{\frac{\iota(T)-k+1}{\sum_{i=k}^T \frac{1}{[\sigma_T^{(i)}]^2}}} \,\middle|\, x \in [\sigma_T^{(k-1)}, \sigma_T^{(k)}] \right\}$ is the solution to the following equation: $x^2 = \frac{\iota(t)}{\sum_{i=1}^t \frac{1}{\max(\sigma_i^2, x^2)}}$. We further have

$$x^2 \leq \frac{\iota(t)}{\sum_{i=1}^t \frac{1}{\sigma_i^2 + x^2}} \leq \frac{\iota(t)}{\frac{t}{x^2 + t^{-1}\sum_{i=1}^t \sigma_i^2}} = \frac{\iota(t)(x^2 + t^{-1}\sum_{i=1}^t \sigma_i^2)}{t}, \tag{4.2}$$

Table 2: Simple and Cumulative Regrets of Algorithm 1 and Weighted-OFUL (Zhou & Gu, 2022). We use $R(T) \leq \sum_{t=1}^{T} \mathrm{SR}(t)$ to upper bound the regret of Algorithm 1. All comparisons in Table 2 are made under the assumption that $\sum_{t=1}^{T} 1/\sigma_t^2 \gg \sum_{i=1}^{\tilde{O}(d)} 1/[\sigma^{(i)}]^2$. For the concrete variance profiles in the table this assumption holds when $T$ is large enough relative to $d$: for the fast-decaying, flat-noise, and many-moderate-spike profiles it is satisfied as soon as $T \gg d$, while for the front-loaded super-precision profile it holds once $T \gg d^{5/4}$.

| Scenario | $\sigma_t^2$ | Simple Regret | | Cumulative Regret | |
|---|---|---|---|---|---|
| | | Algorithm 1 | Weighted OFUL | Algorithm 1 | Weighted OFUL |
| **Fast-Decaying Noise** | $\sigma_t^2 = 1/t^2$ | $\tilde{O}(\frac{d}{T^{3/2}})$ | $\tilde{O}(\frac{d}{T})$ | $\tilde{O}(d)$ | $\tilde{O}(d)$ |
| **Flat Noise** $(1/d)$ | $\sigma_t^2 \equiv 1/d$ | $\tilde{O}(\sqrt{\frac{d}{T}})$ | $\tilde{O}(\sqrt{\frac{d}{T}})$ | $\tilde{O}(\sqrt{dT})$ | $\tilde{O}(\sqrt{dT})$ |
| **Many Moderate Spike** | $\alpha \in (0,1),\ \sigma_t^2 = \begin{cases} x, & t \leq \alpha T, \\ 1, & t > \alpha T, \end{cases}$ with $x = T^{-1/3}$ | $\tilde{O}(\frac{d}{T^{2/3}})$ | $\tilde{O}(\frac{d}{\sqrt{T}})$ | $\tilde{O}(dT^{1/3})$ | $\tilde{O}(d\sqrt{T})$ |
| **Front-Loaded Super-Precision** | $\sigma_t^2 = \begin{cases} \min\{1/2, t^{-2}\}, & t \leq T^{4/5}, \\ 1/2, & t > T^{4/5}, \end{cases}$ | $\tilde{O}(\frac{d}{T^{6/5}})$ | $\tilde{O}(\frac{d}{\sqrt{T}})$ | $\tilde{O}(d)$ | $\tilde{O}(d\sqrt{T})$ |

where the second inequality follows from mean inequality. Rearranging the terms yields $x^2 = \tilde{O}(d\Lambda/t^2)$ when $t = \Omega(d)$. Please refer to Appendix B for detailed derivations.

### 4.3 COMPARISON WITH WEIGHTED OFUL

In this subsection, we present a case study to illustrate the limitations of using the cumulative variance $\Lambda = \sum_{t=1}^{T} \sigma_t^2$ as a measure of statistical complexity in linear bandits with heteroscedastic noise and the potential weakness of existing algorithms that rely on this measure. For simplicity, we assume $\sum_{t=1}^{T} 1/\sigma_t^2 \gg \sum_{i=1}^{\tilde{O}(d)} 1/[\sigma^{(i)}]^2$. Therefore, according to (4.1) and Zhou & Gu (2022),

$$\mathrm{SR}_{\mathrm{Alg\,1}} \asymp d\Big( \sum_{t=1}^{T} \frac{1}{\sigma_t^2} \Big)^{-1/2}, \qquad \mathrm{SR}_{\mathrm{Weighted-OFUL}} \asymp d\frac{\sqrt{\sum_{t=1}^{T} \sigma_t^2}}{T} \tag{4.3}$$

First, by the HM-AM inequality, we have $T/(\sum_t \frac{1}{\sigma_t^2}) \leq \sum_t \sigma_t^2/T$, which leads to a general relationship between the regret bounds of our methods: $\mathrm{SR}_{\mathrm{Alg\,1}} \leq \mathrm{SR}_{\mathrm{Weighted-OFUL}}$ for any sequence $\sigma_t^2$. Therefore, our regret bound is always sharper whenever $\sum_{t=1}^{T} 1/\sigma_t^2 \gg \sum_{i=1}^{\tilde{O}(d)} 1/[\sigma^{(i)}]^2$. In the Table 2, we demonstrate the specific rate of improvement for some special variance sequences. We note that an improvement in simple regret does not necessarily lead to an improvement in cumulative regret. For example, this is evident in fast-decaying noise, as discussed in Remark 3.4.

## 5 STOCHASTIC LINEAR BANDITS WITH FINITE ACTION SPACE

In this section, we consider the special case where the action set $\mathcal{A}$ is finite. We propose a variance-adaptive G-optimal design based exploration strategy and establish a simple regret bound with harmonic-mean dependence which improves over Theorem 4.1 by a factor of $\sqrt{d}$.

### 5.1 VARIANCE-ADAPTIVE G-OPTIMAL DESIGN BASED EXPLORATION

**G-optimal design.** In Algorithm 2, we need to find a nearly $G$-optimal design $\pi \in \Delta(\mathcal{A})$ that maximizes $\log \det V(\pi)$. We first introduce some necessary notations and definitions regarding $D$-optimal and $G$-optimal designs. Let $\pi : \mathcal{A} \to [0,1]$ be a distribution on $\mathcal{A}$ so that $\sum_{\mathbf{a} \in \mathcal{A}} \pi(\mathbf{a}) = 1$. Based on $\pi \in \mathcal{P}(\mathcal{A}_\ell)$, define $V(\pi) \in \mathbb{R}^{d \times d}$ and $g(\pi) \in \mathbb{R}$ as follows $V(\pi) = \sum_{\mathbf{a} \in \mathcal{A}} \pi(\mathbf{a}) \mathbf{a}\mathbf{a}^\top$, $g(\pi) = \max_{\mathbf{a} \in \mathcal{A}} \|\mathbf{a}\|_{V(\pi)^{-1}}^2$. A design $\pi$ is defined as a $G$-optimal design if it minimises $g$. And a design $\pi$ is defined as a $D$-optimal design if it maximises $f(\pi) = \log \det V(\pi)$. The set $\mathrm{Supp}(\pi)$ is sometimes called the core set. The following theorem characterizes the size of the core set and the minimum of $g$ and establishes the equivalence of $G$-optimal and $D$-optimal designs.

---

**Algorithm 2** Variance Adaptive G-Optimal Design (VAGD)

**Require:** $\mathcal{A} \subset \mathbb{R}^d$, $\delta$.
1: Find nearly $G$-optimal design $\pi \in \Delta(\mathcal{A})$ with $|\mathrm{supp}(\pi)| \leq 4d \log \log d + 16$ as described in Theorem 5.2 that minimizes

$$\max_{\mathbf{a} \in \mathcal{A}} \|\mathbf{a}\|_{V(\pi)^{-1}} \text{ subject to } \sum_{\mathbf{a} \in \mathcal{A}} \pi(\mathbf{a}) = 1.$$

2: Let $\mathcal{T}_0(\mathbf{a}) \leftarrow \varnothing$ for all $\mathbf{a} \in \mathcal{A}$.
3: **for** $t = 1, \ldots, T$ **do**
4:     Pull the action $\mathbf{a}_t := \mathrm{argmin}_{\mathbf{a} \in \mathcal{A}} \sum_{\tau \in \mathcal{T}(\mathbf{a})} \frac{1}{\sigma_\tau^2 \cdot \pi(\mathbf{a})}$
5:     Observe reward $r_t$ and variance $\sigma_t$.
6:     Update the set $\mathcal{T}_t(\mathbf{a}_t) \leftarrow \mathcal{T}_{t-1}(\mathbf{a}_t) \cup \{t\}$ and $\mathcal{T}_t(\mathbf{a}) \leftarrow \mathcal{T}_{t-1}(\mathbf{a})$ for all $\mathbf{a} \neq \mathbf{a}_t$.
7: **end for**
8: Outout $\mathbf{a}_{T+1} = \mathrm{argmax}_{\mathbf{a} \in \mathcal{A}} \langle \hat{\boldsymbol{\theta}}_T, \mathbf{a} \rangle$ where $\hat{\boldsymbol{\theta}}_T = V_T^{-1} \sum_{t=1}^T \sigma_t^{-2} r_t \mathbf{a}_t$ and $V_T = I + \sum_{t=1}^T \sigma_t^{-2} \mathbf{a}_t \mathbf{a}_t^\top$.

---

**Theorem 5.1.** (Lattimore & Szepesvári, 2020, Kiefer-Wolfowitz) Assume that $\mathcal{A} \subset \mathbb{R}^d$ is compact and $\mathrm{span}(\mathcal{A}) = \mathbb{R}^d$. The following are equivalent: (a) $\pi^*$ is a minimiser of $g$; (b) $\pi^*$ is a maximiser of $f(\pi) = \log \det V(\pi)$; (c) $g(\pi^*) = d$.

Furthermore, there exists a minimiser $\pi^*$ of $g$ such that $|\mathrm{Supp}(\pi^*)| \leq d(d+1)/2$.

However, the core set size of the $G$-optimal design given by Theorem 5.1 is at most $d(d+1)/2$, which may cause additional overhead in our variance-adaptive algorithm. To address this issue, we can find an approximate $G$-optimal design with a smaller core set size using the following theorem.

**Theorem 5.2** (Lattimore et al. 2020). Suppose that $\mathcal{A} \subset \mathbb{R}^d$ is compact and $\mathrm{span}(\mathcal{A}) = \mathbb{R}^d$. There exists a probability distribution $\pi \in \Delta(\mathcal{A})$ such that $g(\pi) \leq 2d$ and the cardinality of the core set of $\pi$ is at most $4d \log \log d + 16$.

**Adaptive arm selection.** After obtaining the approximate $G$-optimal design $\pi$, we use it to guide the arm selection process. Unlike traditional $G$-optimal design-based algorithms, which pull arms according to the fixed distribution $\pi$, our algorithm adaptively selects arms based on the observed variances $\sigma_t$. Specifically, at each round $t$, we choose the arm $\mathbf{a}_t$ that has been pulled the fewest times relative to its probability under $\pi$, weighted by the inverse of the observed variance. This adaptive strategy prevents over-exploration caused by the unpredictable and heteroscedastic nature of noise and ensures that we collect sufficient information from all arms in the core set of $\pi$.

**Weighted least-squares estimator.** Inspired by Zhou et al. (2021), we use a variance-weighted least-squares estimator to estimate the unknown parameter $\boldsymbol{\theta}^*$. Specifically, after $T$ rounds of exploration, we compute the estimator $\hat{\boldsymbol{\theta}}_T$ as follows:

$$\hat{\boldsymbol{\theta}}_T = V_T^{-1} \sum_{t=1}^T \sigma_t^{-2} r_t \mathbf{a}_t, \quad V_T = I + \sum_{t=1}^T \sigma_t^{-2} \mathbf{a}_t \mathbf{a}_t^\top.$$

Under the finite action space regime, we show that this estimator achieves a tighter confidence bound compared to the general case, replacing the $\sqrt{d}$ factor with $\sqrt{\log(|\mathcal{A}|)}$ in the confidence radius.

Finally, we recommend the action $\mathbf{a}_{T+1}$ that maximizes the estimated reward based on $\hat{\boldsymbol{\theta}}_T$.

### 5.2 Simple Regret Bound for VAGD

**Theorem 5.3** (Simple Regret of Algorithm 2). Suppose that $\mathcal{A} \subset \mathbb{R}^d$ is compact and $\mathrm{span}(\mathcal{A}) = \mathbb{R}^d$. If we follow Algorithm 2, then it holds that with probability at least $1 - \delta$,

$$\langle \boldsymbol{\theta}^*, \mathbf{a}^* \rangle - \langle \boldsymbol{\theta}^*, \mathbf{a}_{T+1} \rangle \leq 2 \sqrt{d \log(|\mathcal{A}|/\delta) \Big/ \Big[ \sum_{t=1}^T \frac{1}{\sigma_t^2} - \sum_{i=1}^{4d \log \log d + 16} \frac{1}{[\sigma_T^{(i)}]^2} \Big]}.$$

**Remark 5.4.** Theorem 5.3 establishes a simple regret bound for Algorithm 2 that depends on the harmonic mean of the noise variances $\sigma_t^2$, while improving the dependence on the dimension $d$ compared to Theorem 4.1. In particular, the $\sqrt{d}$ factor in the numerator is replaced by $\sqrt{\log(|\mathcal{A}|)}$, which can be substantially smaller when the action set $\mathcal{A}$ is finite and of moderate size. This improvement is obtained by exploiting the finite action space structure and employing a variance-adaptive G-optimal design exploration strategy. Consequently, Algorithm 2 is especially effective in settings with limited action sets, enabling more efficient exploration and improved simple regret performance.

## 6 LOWER BOUND

In this section, we establish a lower bound for the simple regret in linear bandits with heteroscedastic noise. Our lower bound nearly matches the upper bound in Theorem 4.1 up to logarithmic factors, demonstrating the optimality of our proposed algorithm.

**Theorem 6.1** (Instance-dependent lower bound.). For any $d \geq 2$ and $T \geq 1$, and any algorithm $\mathcal{A}$, there exists a linear bandit instance with heteroscedastic Gaussian noise satisfying our assumptions such that the simple regret is lower bounded as follows:

$$\mathbb{E}[\text{SR}(T)] \geq \frac{3}{16} d \cdot \left( \sum_{t=1}^{T} \frac{1}{\sigma_t^2} \right)^{-1/2}.$$

**Remark 6.2.** Theorem 6.1 establishes an variance-sequence-dependent lower bound for the simple regret in linear bandits with heteroscedastic noise. This lower bound matches the upper bound in Theorem 4.1 up to logarithmic factors, indicating that our proposed algorithm is nearly optimal in terms of its dependence on the harmonic mean of the variances $\sigma_t^2$. This result highlights the fundamental difficulty of the best-arm identification problem in linear bandits with heteroscedastic noise and underscores the effectiveness of our variance-adaptive approach. In contrast, in Table 1, the worst-case lower bound established in previous studies (He & Gu, 2025; Jia et al., 2024) is derived by constructing an instance where all variances are equal, which does not capture the complexity of heteroscedastic linear bandits, especially when the variances vary significantly across actions and time steps.

## 7 CONCLUSION AND FUTURE WORK

In this paper, we study the sample complexity of stochastic linear bandits with heteroscedastic noise under a fixed action set. We propose a variance-adaptive algorithm that achieves a nearly instance-optimal simple regret bound, characterized by the harmonic mean of the noise variances. We further establish a nearly matching lower bound, demonstrating the optimality of our algorithm. Together, these results provide a comprehensive characterization of the statistical complexity of linear bandits with heteroscedastic noise.

There are several promising directions for future work. First, one could consider settings where the context is not fixed but instead sampled from an unknown distribution. Second, it would be natural to extend our results to the case where the noise variances are unknown and must be estimated from data. Third, in reinforcement learning, the variance of the noise is governed by the transition dynamics, which can themselves be estimated from historical data. Extending our results to Markov decision processes using such estimated variances would be an interesting direction to explore.

ACKNOWLEDGMENT

We thank the anonymous reviewers and area chair for their helpful comments. HZ and QG are supported in part by the National Science Foundation DMS-2323113 and IIS-2403400. HZ is also supported in part by Amazon PhD Fellowship. T. Jin and V. Y. F. Tan are supported by a Singapore Ministry of Education (MOE) AcRF Tier 2 grant under grant number A-8004062-00-00. WW and PX are supported in part by the National Science Foundation (DMS-2323112) and the Whitehead Scholars Program at the Duke University School of Medicine. The views and conclusions contained in this paper are those of the authors and should not be interpreted as representing any funding agencies.

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

## A  LLM USAGE

We used an LLM only for grammatical and stylistic polishing of the manuscript. No research ideas or results were generated by the LLM. The authors wrote and verified all technical content.

## B  PROOF OF THEOREM 4.1

**Lemma B.1** (Matrix Inversion Lemma, Harville (1998)). *For any invertible matrix $A \in \mathbb{R}^{d \times d}$, vector $\mathbf{u}, \mathbf{v} \in \mathbb{R}^d$, it holds that*

$$(A + \mathbf{u}\mathbf{v}^\top)^{-1} = A^{-1} - \frac{A^{-1}\mathbf{u}\mathbf{v}^\top A^{-1}}{1 + \mathbf{v}^\top A^{-1}\mathbf{u}}.$$

**Lemma B.2** (Elliptical Potential Lemma, Abbasi-Yadkori et al. (2011)). *For any sequence of vectors $\{\mathbf{x}_t\}_{t=1}^T \subset \mathbb{R}^d$, let $V_0 = \lambda \mathbf{I}$ for some $\lambda > 0$ and $V_t = V_{t-1} + \mathbf{x}_t \mathbf{x}_t^\top$ for $t \geq 1$. If $\|\mathbf{x}_t\|_2 \leq L$ for all $t$, then we have*

$$\sum_{t=1}^T \min\{1, \|\mathbf{x}_t\|_{V_{t-1}^{-1}}^2\} \leq 2d \log \frac{\lambda + TL^2}{d\lambda}.$$

**Lemma B.3.** *With probability at least $1 - \delta$, it holds for all $t \in [T]$ that*

$$\|\hat{\boldsymbol{\theta}}_t - \boldsymbol{\theta}^*\|_{V_t} \leq \beta_t := 2\sqrt{\lambda} + 16\sqrt{\log(4t^2/\delta) \cdot d \log \frac{d\lambda + t\sigma_{\min}^{-2}}{d\lambda}}.$$

*Proof.* The proof follows from the standard analysis of OFUL (Abbasi-Yadkori et al., 2011) with variance-weighted updates.

We have

$$\|\hat{\boldsymbol{\theta}}_t - \boldsymbol{\theta}^*\|_{V_t}^2 = (\hat{\boldsymbol{\theta}}_t - \boldsymbol{\theta}^*)^\top V_t (\hat{\boldsymbol{\theta}}_t - \boldsymbol{\theta}^*)$$

$$= \left(\sum_{s=1}^t \sigma_s^{-2}\mathbf{a}_s r_s - \sum_{s=1}^t \sigma_s^{-2}\mathbf{a}_t\mathbf{a}_t^\top\boldsymbol{\theta}^* - \lambda\boldsymbol{\theta}^*\right)^\top V_t^{-1} \cdot V_t \cdot V_t^{-1} \cdot \left(\sum_{s=1}^t \sigma_s^{-2}\mathbf{a}_s r_s - \sum_{s=1}^t \sigma_s^{-2}\mathbf{a}_t\mathbf{a}_t^\top\boldsymbol{\theta}^* - \lambda\boldsymbol{\theta}^*\right)$$

$$= \left(\sum_{s=1}^t \sigma_s^{-2}\mathbf{a}_s\eta_s - \lambda\boldsymbol{\theta}^*\right) V_t^{-1} \left(\sum_{s=1}^t \sigma_s^{-2}\mathbf{a}_s\eta_s - \lambda\boldsymbol{\theta}^*\right)$$

$$\leq 2\lambda^2\|\boldsymbol{\theta}^*\|_{V_t^{-1}}^2 + 2\underbrace{\left(\sum_{s=1}^t \sigma_s^{-2}\mathbf{a}_s\eta_s\right)^\top V_t^{-1} \left(\sum_{s=1}^t \sigma_s^{-2}\mathbf{a}_s\eta_s\right)}_{I_{0,t}}, \tag{B.1}$$

where the second equality follows from the definition of $\hat{\boldsymbol{\theta}}_t$, the third equality follows from the definition of $r_s$ and $\eta_s$, the inequality follows from Young's inequality. To further bound $I_{0,t}$, we introduce the following notation:

$$\mathbf{d}_0 = 0, \quad \mathbf{d}_t = \sum_{s=1}^t \sigma_s^{-2}\mathbf{a}_s\eta_s,$$

$$\mathcal{I}_t = \mathbb{1}(0 \leq s \leq t, I_{0,s} \leq \gamma_s), \quad \gamma_s := 64\log(4s^2/\delta) \cdot d \log \frac{d\lambda + s\sigma_{\min}^{-2}}{d\lambda}.$$

Decomposing $I_{0,t}$ into a martingale difference sequence, we have

$$I_{0,t} = \mathbf{d}_{t-1}^\top V_t^{-1}\mathbf{d}_{t-1} + 2\sigma_t^{-2}\eta_t\mathbf{a}_t^\top V_t^{-1}\mathbf{d}_{t-1} + \sigma_t^{-4}\eta_t^2\mathbf{a}_t^\top V_t^{-1}\mathbf{a}_t$$

$$\leq I_{0,t-1} + 2\underbrace{\sigma_t^{-2}\eta_t\mathbf{a}_t^\top V_t^{-1}\mathbf{d}_{t-1}}_{I_{1,t}} + \underbrace{\sigma_t^{-4}\eta_t^2\mathbf{a}_t^\top V_t^{-1}\mathbf{a}_t}_{I_{2,t}}. \tag{B.2}$$

From the matrix inversion lemma (Lemma B.1), we have

$$
\begin{aligned}
I_{1,t} &= \sigma_t^{-2} \eta_t \mathbf{a}_t^\top \left( V_{t-1}^{-1} - \frac{\sigma_t^{-2} V_{t-1}^{-1} \mathbf{a}_t \mathbf{a}_t^\top V_{t-1}^{-1}}{1 + \sigma_t^{-2} \mathbf{a}_t^\top V_{t-1}^{-1} \mathbf{a}_t} \right) \mathbf{d}_{t-1} \\
&= \sigma_t^{-2} \eta_t \left( \mathbf{a}_t V_{t-1}^{-1} \mathbf{d}_{t-1} - \frac{\sigma_t^{-2} \|\mathbf{a}_t\|_{V_{t-1}^{-1}}^2 \mathbf{a}_t V_{t-1}^{-1} \mathbf{d}_{t-1}}{1 + \sigma_t^{-2} \|\mathbf{a}_t\|_{V_{t-1}^{-1}}^2} \right) \\
&= \sigma_t^{-2} \eta_t \frac{\mathbf{a}_t V_{t-1}^{-1} \mathbf{d}_{t-1}}{1 + \sigma_t^{-2} \|\mathbf{a}_t\|_{V_{t-1}^{-1}}^2}.
\end{aligned}
$$

Based on our assumption on the noise $\eta_t$, $I_{1,t} \cdot \mathcal{I}_t$ is also a sub-Gaussian random variable with variance proxy bounded by

$$
\sigma_t^{-2} \frac{\|\mathbf{a}_t\|_{V_{t-1}^{-1}}^2 \|\mathbf{d}_{t-1}\|_{V_{t-1}^{-1}}^2}{(1 + \sigma_t^{-2} \|\mathbf{a}_t\|_{V_{t-1}^{-1}}^2)^2} \cdot \mathcal{I}_t.
$$

Adding $I_{1,s} \cdot \mathcal{I}_s$ up to $t$ and using Lemma D.3, we have with probability at least $1 - \delta/2$,

$$
\begin{aligned}
\sum_{s=1}^t I_{1,s} \cdot \mathcal{I}_s &\leq \sqrt{2 \log(2/\delta) \sum_{s=1}^t \sigma_s^{-2} \frac{\|\mathbf{a}_s\|_{V_{s-1}^{-1}}^2 \|\mathbf{d}_{s-1}\|_{V_{s-1}^{-1}}^2}{(1 + \sigma_s^{-2} \|\mathbf{a}_s\|_{V_{s-1}^{-1}}^2)^2} \cdot \mathcal{I}_s} \\
&\leq \sqrt{2 \log(2/\delta) \sum_{s=1}^t \min\{1, \|\sigma_s^{-1} \mathbf{a}_s\|_{V_{s-1}^{-1}}^2\} \cdot \gamma_t} \\
&\leq \sqrt{2\gamma_t \log(2/\delta) \cdot 2d \log \frac{d\lambda + t\sigma_{\min}^{-2}}{d\lambda}} \\
&\leq \frac{1}{4}\gamma_t + 8 \log(2/\delta) \cdot d \log \frac{d\lambda + t\sigma_{\min}^{-2}}{d\lambda},
\end{aligned} \tag{B.3}
$$

where the second inequality follows from the definition of $\mathcal{I}_s$ and the fact that $\frac{\sigma_s^{-2} \|\mathbf{a}_s\|_{V_{s-1}^{-1}}^2}{(1 + \sigma_s^{-2} \|\mathbf{a}_s\|_{V_{s-1}^{-1}}^2)^2} \leq 1$, the third inequality follows from Lemma B.2, and the last inequality follows from Young's inequality.

Using union bound over all $t \geq 1$, we have with probability at least $1 - \delta/2$,

$$
\sum_{s=1}^t I_{1,s} \cdot \mathcal{I}_s \leq \frac{1}{4}\gamma_t + 8 \log(4t^2/\delta) \cdot d \log \frac{d\lambda + t\sigma_{\min}^{-2}}{d\lambda}
$$

for all $t \geq 1$.

For the second term $I_{2,t}$, it follows from the matrix inversion lemma (Lemma B.1) that

$$
\begin{aligned}
I_{2,t} &= \sigma_t^{-4} \eta_t^2 \mathbf{a}_t^\top \left( V_{t-1}^{-1} - \frac{\sigma_t^{-2} V_{t-1}^{-1} \mathbf{a}_t \mathbf{a}_t^\top V_{t-1}^{-1}}{1 + \sigma_t^{-2} \mathbf{a}_t^\top V_{t-1}^{-1} \mathbf{a}_t} \right) \mathbf{a}_t \\
&= \sigma_t^{-4} \eta_t^2 \left( \|\mathbf{a}_t\|_{V_{t-1}^{-1}}^2 - \frac{\sigma_t^{-2} \|\mathbf{a}_t\|_{V_{t-1}^{-1}}^4}{1 + \sigma_t^{-2} \|\mathbf{a}_t\|_{V_{t-1}^{-1}}^2} \right) \\
&= \sigma_t^{-4} \eta_t^2 \frac{\|\mathbf{a}_t\|_{V_{t-1}^{-1}}^2}{1 + \sigma_t^{-2} \|\mathbf{a}_t\|_{V_{t-1}^{-1}}^2},
\end{aligned}
$$

Using union bound over $t \geq 1$, we have with probability at least $1 - \delta/2$,

$$I_{2,t} \leq \sigma_t^{-4} \sigma_t^2 \log(4t^2/\delta) \frac{\|\mathbf{a}_t\|_{V_{t-1}^{-1}}^2}{1 + \sigma_t^{-2}\|\mathbf{a}_t\|_{V_{t-1}^{-1}}^2}$$

for all $t \geq 1$.

Thus, for any $t \geq 1$,

$$\sum_{s=1}^{t} I_{2,s} \leq \sum_{s=1}^{t} \sigma_s^{-2} \log(4s^2/\delta) \frac{\|\mathbf{a}_s\|_{V_{s-1}^{-1}}^2}{1 + \sigma_s^{-2}\|\mathbf{a}_s\|_{V_{s-1}^{-1}}^2}$$

$$\leq \log(4t^2/\delta) \sum_{s=1}^{t} \min\{1, \|\sigma_s^{-1}\mathbf{a}_s\|_{V_{s-1}^{-1}}^2\}$$

$$\leq 2\log(4t^2/\delta) \cdot d \log \frac{d\lambda + t\sigma_{\min}^{-2}}{d\lambda}, \tag{B.4}$$

where the last inequality follows from Lemma B.2.

Substituting (B.3) and (B.4) into (B.2), and using induction on $t$, we have with probability at least $1 - \delta$,

$$I_{0,t} \leq \gamma_t := 64\log(4t^2/\delta) \cdot d \log \frac{d\lambda + t\sigma_{\min}^{-2}}{d\lambda},$$

which further implies that

$$\|\hat{\boldsymbol{\theta}}_t - \boldsymbol{\theta}^*\|_{V_t}^2 \leq 2\lambda^2\|\boldsymbol{\theta}^*\|_{V_t^{-1}}^2 + 2I_{0,t} \leq 2\lambda + 256\log(4t^2/\delta) \cdot d \log \frac{d\lambda + t\sigma_{\min}^{-2}}{d\lambda}.$$

$$\square$$

**Lemma B.4.** If we follow Algorithm 1 to choose the action $\mathbf{a}_t$, then it holds for any $t \in [T]$ that

$$\|\mathbf{a}_t\|_{V_{t-1}^{-1}}^2 \leq \min_{1 \leq k \leq t+1}\left\{ x^2 = \frac{\iota(t) - k + 1}{\sum_{i=k}^{t} \frac{1}{[\sigma_t^{(i)}]^2}} \middle| [\sigma_t^{(i)}]^2 \text{ is the } i\text{-th smallest element in } \{\sigma_\tau^2\}_{\tau=1}^{t}, \right.$$

$$\left. x \in [\sigma_t^{(k-1)}, \sigma_t^{(k)}]\right\},$$

where $\iota(t) = 2d\log\left(\frac{d + \sum_{\tau \in [t]} \sigma_\tau^{-2}}{d}\right)$.

*Proof.* When $x \in [0, 1]$, $x \leq 2\log(1 + x)$, which further indicates that

$$\sum_{\tau \in [t]} \min\left\{1, \frac{1}{\sigma_\tau^2}\|\mathbf{a}_\tau\|_{V_{\tau-1}^{-1}}^2\right\} \leq 2 \sum_{\tau \in [t]} \log\left(1 + \frac{1}{\sigma_\tau^2}\|\mathbf{a}_\tau\|_{V_{\tau-1}^{-1}}^2\right)$$

$$\leq 2\log \frac{\det(V_\tau)}{\det(V_0)}$$

$$\leq 2d\log\left(\frac{d + \sum_{\tau \in [t]} \sigma_\tau^{-2}}{d}\right). \tag{B.5}$$

Let

$$\iota(t) := 2d\log\left(\frac{d + \sum_{\tau \in [t]} \sigma_\tau^{-2}}{d}\right).$$

Note that for $i \leq t$,

$$\|\mathbf{a}_t\|_{V_{t-1}^{-1}} \leq \|\mathbf{a}_t\|_{V_{i-1}^{-1}} \leq \|\mathbf{a}_i\|_{V_{i-1}^{-1}}$$

due to the fact that $V_{t-1} \succeq V_{i-1}$ and the definition of $\mathbf{a}_i$ in Algorithm 1. Therefore, following inequality (B.5), we obtain that

$$\sum_{\tau \in [t]} \min\left\{1, \frac{1}{\sigma_\tau^2}\|\mathbf{a}_t\|^2_{V_{t-1}^{-1}}\right\} \leq \sum_{\tau \in [t]} \min\left\{1, \frac{1}{\sigma_\tau^2}\|\mathbf{a}_\tau\|^2_{V_{\tau-1}^{-1}}\right\} \leq \iota(t). \tag{B.6}$$

As LHS of (B.6) is strictly increasing with respect to $\|\mathbf{a}_t\|_{V_{t-1}^{-1}}$ as long as $\iota(t) < t$, we can derive a bound for $\|\mathbf{a}_t\|_{V_{t-1}^{-1}}$ by solving

$$\sum_{\tau \in [t]} \min\left\{1, \frac{1}{\sigma_\tau^2}x^2\right\} = \iota(t). \tag{B.7}$$

To solve (B.7), we define a sorted sequence of $\{\sigma_i\}_{i=1}^t$ in increasing order, denoted as

$$\sigma_t^{(1)} \leq \sigma_t^{(2)} \leq \cdots \leq \sigma_t^{(t)}.$$

Let $\sigma_t^{(0)} := 0$. Suppose that $x \in [\sigma_t^{(k-1)}, \sigma_t^{(k)}]$ for some $k \in [t]$. Then for $i < k$, we have

$$\min\left\{1, \frac{1}{[\sigma_t^{(i)}]^2}x^2\right\} = 1;$$

for $i \geq k$, we have

$$\min\left\{1, \frac{1}{[\sigma_t^{(i)}]^2}x^2\right\} = \frac{x^2}{[\sigma_t^{(i)}]^2}.$$

After rewriting the LHS of the above equality, we have

$$k - 1 + \sum_{i=k}^t \frac{x^2}{[\sigma_t^{(i)}]^2} = \iota(t).$$

We can then rearrange the above inequality to obtain

$$\|\mathbf{a}_t\|^2_{V_{t-1}^{-1}} \leq x^2,$$

where

$$x^2 := \min_{1 \leq k \leq t+1}\left\{\frac{\iota(t) - k + 1}{\sum_{i=k}^t \frac{1}{[\sigma_t^{(i)}]^2}}\, \middle|\, [\sigma_t^{(k-1)}]^2 \leq \frac{\iota(t) - k + 1}{\sum_{i=k}^t \frac{1}{[\sigma_t^{(i)}]^2}} \leq [\sigma_t^{(k)}]^2\right\}.$$

This completes the proof. □

**Remark B.5.** In the previous proof, $x^2$ is the solution for the implicit equation (B.7). We can also rewrite it as

$$sx^2 \sum_{i=1}^t \frac{1}{\max(\sigma_i^2, x^2)} = \iota(t),$$

which implies that

$$x^2 = \frac{\iota(t)}{\sum_{i=1}^t \frac{1}{\max(\sigma_i^2, x^2)}}.$$

We further have

$$x^2 \leq \frac{\iota(t)}{\sum_{i=1}^t \frac{1}{\sigma_i^2 + x^2}} \leq \frac{\iota(t)}{\frac{t}{x^2 + t^{-1}\sum_{i=1}^t \sigma_i^2}} = \frac{\iota(t)(x^2 + t^{-1}\sum_{i=1}^t \sigma_i^2)}{t}.$$

Rearranging the above inequality, we obtain

$$x^2 \leq \frac{\iota(t)\sum_{i=1}^t \sigma_i^2}{t[t - \iota(t)]} = \tilde{O}\left(\frac{d\sum_{i=1}^t \sigma_i^2}{t^2}\right)$$

when $t = \Omega(d)$.

**Theorem B.6** (Simple Regret of VAEE, restatement of Theorem 4.1). If we set $\lambda = 1$ and $\beta_t = 2\sqrt{\lambda} + 16\sqrt{\log(4t^2/\delta) \cdot d \log \frac{d\lambda + t\sigma_{\min}^{-2}}{d\lambda}}$ in Algorithm 1, then with probability at least $1 - \delta$, the simple regret of Algorithm 1 is bounded as

$$\mathrm{SR}(T) = \tilde{O}(\sqrt{d}) \min_{1 \leq k \leq T+1} \left\{ x = \sqrt{\frac{\iota(T) - k + 1}{\sum_{i=k}^{T} \frac{1}{[\sigma_T^{(i)}]^2}}} \middle| [\sigma_T^{(i)}]^2 \text{ is the } i\text{-th smallest element in } \{\sigma_\tau^2\}_{\tau=1}^T, \right.$$

$$\left. x \in [\sigma_T^{(k-1)}, \sigma_T^{(k)}] \right\},$$

where $\iota(T) = 2d \log\left(\frac{d + \sum_{\tau \in [T]} \sigma_\tau^{-2}}{d}\right)$.

*Proof.* By Lemma B.3, with probability at least $1 - \delta$, it holds for all $t \in [T]$ that $\boldsymbol{\theta}^* \in \mathcal{C}_t := \{\boldsymbol{\theta} \in \mathbb{R}^d : \|\hat{\boldsymbol{\theta}}_t - \boldsymbol{\theta}\|_{V_t} \leq \beta_t\}$. In the following, we condition on the event that $\boldsymbol{\theta}^* \in \mathcal{C}_t$ for all $t \in [T]$.

With the conditioned event, we can show by induction that for any $t \in [T]$, $\mathbf{a}^* \in \mathcal{A}_t$:

$$\max_{\boldsymbol{\theta} \in \mathcal{C}_{t-1}} \langle \boldsymbol{\theta}, \mathbf{a}^* \rangle \geq \langle \boldsymbol{\theta}^*, \mathbf{a}^* \rangle \geq \max_{\mathbf{a} \in \mathcal{A}_{t-1}} \langle \boldsymbol{\theta}^*, \mathbf{a} \rangle \geq \max_{\mathbf{a} \in \mathcal{A}_{t-1}} \min_{\boldsymbol{\theta} \in \mathcal{C}_{t-1}} \langle \boldsymbol{\theta}, \mathbf{a} \rangle,$$

where the first inequality follows from the fact that $\boldsymbol{\theta}^* \in \mathcal{C}_{t-1}$, the second inequality follows from the definition of $\mathbf{a}^*$, and the last inequality follows from the definition of $\mathcal{A}_{t-1}$.

Let $\mathbf{a}^* = \mathrm{argmax}_{\mathbf{a} \in \mathcal{A}} \langle \boldsymbol{\theta}^*, \mathbf{a} \rangle$ be the optimal action. By the definition of simple regret, we have

$$\mathrm{SR}(T) = \langle \boldsymbol{\theta}^*, \mathbf{a}^* - \mathbf{a}_T \rangle$$
$$\leq \max_{\mathbf{a} \in \mathcal{A}_T} \max_{\boldsymbol{\theta} \in \mathcal{C}_{T-1}} \langle \boldsymbol{\theta}, \mathbf{a} \rangle - \langle \boldsymbol{\theta}^*, \mathbf{a}_T \rangle$$
$$\leq \max_{\mathbf{a} \in \mathcal{A}_T} \langle \hat{\boldsymbol{\theta}}_{T-1}, \mathbf{a} \rangle + \beta_T \|\mathbf{a}_T\|_{V_{T-1}^{-1}} - \langle \hat{\boldsymbol{\theta}}_{T-1}, \mathbf{a}_T \rangle + \beta_T \|\mathbf{a}_T\|_{V_{T-1}^{-1}},$$

where the first inequality follows from the event that $\boldsymbol{\theta}^* \in \mathcal{C}_t$ for all $t \in [T]$, and the fact that $\mathbf{a}^* \in \mathcal{A}_T$, the second inequality follows from the definition of $\mathcal{C}_{T-1}$ and the definition of $\mathbf{a}_T$ in Algorithm 1.

Then it suffices to bound $\max_{\mathbf{a} \in \mathcal{A}_T} \langle \hat{\boldsymbol{\theta}}_{T-1}, \mathbf{a} \rangle - \langle \hat{\boldsymbol{\theta}}_{T-1}, \mathbf{a}_T \rangle$. Since $\mathbf{a}_T \in \mathcal{A}_T$, it is guaranteed that there exists $\boldsymbol{\theta}_T' \in \mathcal{C}_{T-1}$ such that

$$\langle \boldsymbol{\theta}_T', \mathbf{a}_T \rangle - \max_{\mathbf{a} \in \mathcal{A}_T} \min_{\boldsymbol{\theta} \in \mathcal{C}_{T-1}} \langle \boldsymbol{\theta}, \mathbf{a} \rangle \geq 0,$$

which further implies that

$$\max_{\mathbf{a} \in \mathcal{A}_T} \langle \hat{\boldsymbol{\theta}}_{T-1}, \mathbf{a} \rangle - \langle \hat{\boldsymbol{\theta}}_{T-1}, \mathbf{a}_T \rangle = \max_{\mathbf{a} \in \mathcal{A}_T} \langle \hat{\boldsymbol{\theta}}_{T-1}, \mathbf{a} \rangle - \langle \boldsymbol{\theta}_T', \mathbf{a}_T \rangle + \langle \boldsymbol{\theta}_T' - \hat{\boldsymbol{\theta}}_{T-1}, \mathbf{a}_T \rangle$$
$$\leq \max_{\mathbf{a} \in \mathcal{A}_T} \langle \hat{\boldsymbol{\theta}}_{T-1}, \mathbf{a} \rangle - \langle \boldsymbol{\theta}_T', \mathbf{a}_T \rangle + \beta_T \|\mathbf{a}_T\|_{V_{T-1}^{-1}}$$
$$\leq 3\beta_T \max_{\mathbf{a} \in \mathcal{A}_T} \|\mathbf{a}\|_{V_{T-1}^{-1}} = 3\beta_T \|\mathbf{a}_T\|_{V_{T-1}^{-1}},$$

where the last inequality holds because $\max_{\mathbf{a} \in \mathcal{A}_T} \langle \hat{\boldsymbol{\theta}}_{T-1}, \mathbf{a} \rangle - \max_{\mathbf{a} \in \mathcal{A}_T} \min_{\boldsymbol{\theta} \in \mathcal{C}_{T-1}} \langle \boldsymbol{\theta}, \mathbf{a} \rangle \leq 2\beta_T \max_{\mathbf{a} \in \mathcal{A}_T} \|\mathbf{a}\|_{V_{T-1}^{-1}}$ based on the definition of $\mathcal{C}_{T-1}$ and the last equality follows from the action selection rule in Algorithm 1.

Hence, the simple regret of Algorithm 1 is bounded as

$$\mathrm{SR}(T) \leq 5\beta_T \|\mathbf{a}_T\|_{V_{T-1}^{-1}}$$

$$\leq \tilde{O}(\sqrt{d}) \min_{1 \leq k \leq T+1} \left\{ x = \sqrt{\frac{\iota(T) - k + 1}{\sum_{i=k}^{T} \frac{1}{[\sigma_T^{(i)}]^2}}} \middle| [\sigma_T^{(i)}]^2 \text{ is the } i\text{-th smallest element in } \{\sigma_\tau^2\}_{\tau=1}^T, \right.$$

$$\left. x \in [\sigma_T^{(k-1)}, \sigma_T^{(k)}] \right\},$$

where the last inequality follows from the choice of $\beta_T$ and the result in the previous lemma. $\qquad\square$

## C PROOF OF THEOREM 6.1

The proofs of our lower bound require the following Lemmas, which is standard technique used for lower bound.

**Lemma C.1** (Le Cam two-point method (Le Cam, 1986)). Let $P$ and $Q$ be probability measures on the same measurable space, and let $\phi : \Omega \to \{0, 1\}$ be any (possibly randomized) test. Then

$$\frac{1}{2}\Big(P\{\phi = 1\} + Q\{\phi = 0\}\Big) \geq \frac{1}{2}\Big(1 - \delta_{\mathrm{TV}}(P, Q)\Big),$$

where $\delta_{\mathrm{TV}}(P, Q) = \sup_A |P(A) - Q(A)|$ is total variation distance. Moreover, by Pinsker's inequality,

$$\delta_{\mathrm{TV}}(P, Q) \leq \sqrt{\tfrac{1}{2}\,\mathrm{KL}(P\|Q)}.$$

Hence the average error of any test is bounded below by

$$\frac{1}{2}\Big(P\{\phi = 1\} + Q\{\phi = 0\}\Big) \geq \tfrac{1}{2}\Big(1 - \sqrt{\tfrac{1}{2}\,\mathrm{KL}(P\|Q)}\Big).$$

**Lemma C.2** (Pinsker's inequality (Pinsker, 1964)). For any probability measures $P, Q$,

$$\delta_{\mathrm{TV}}(P, Q) \leq \sqrt{\tfrac{1}{2}\,\mathrm{KL}(P\|Q)}.$$

**Lemma C.3** (Yao's minimax principle (Yao, 1977)). Let $\Pi$ be the set of deterministic algorithms (measurable decision rules), $\mathcal{P}$ a family of instances with loss $L(\pi, \boldsymbol{\theta})$ and let $\mathcal{D}$ be distributions over $\mathcal{P}$. Then

$$\inf_{\pi \in \Pi} \sup_{\boldsymbol{\theta} \in \mathcal{P}} \mathbb{E}_{\boldsymbol{\theta}}\big[L(\pi, \boldsymbol{\theta})\big] \geq \sup_{\mu \in \mathcal{D}} \inf_{\pi \in \Pi} \mathbb{E}_{\boldsymbol{\theta} \sim \mu} \mathbb{E}_{\boldsymbol{\theta}}\big[L(\pi, \boldsymbol{\theta})\big].$$

In other words, the worst-case risk of the best deterministic algorithm is at least the Bayes risk under any prior $\mu$.

Now, we are ready to prove our lower bound.

*Proof of Theorem 6.1.* **Step 1 (per-coordinate two-point divergence).** Let action set $\mathcal{A} = \{-1, 1\}^d$. The unknown parameter belongs to

$$\Theta = \{-c, +c\}^d \quad \text{for some } c > 0.$$

The optimal arm for $\boldsymbol{\theta}$ is $\boldsymbol{a}^*(\boldsymbol{\theta}) = \mathrm{sign}(\boldsymbol{\theta})$. Since each coordinate mistake costs $2c$, the *simple regret* is

$$\mathrm{SR}(T) = \mathbb{E}_{\boldsymbol{\theta}}\big[\langle \boldsymbol{\theta}, \boldsymbol{a}^*(\boldsymbol{\theta})\rangle - \langle \boldsymbol{\theta}, \hat{\boldsymbol{a}}_T\rangle\big] = 2c \cdot \mathbb{E}_{\boldsymbol{\theta}}\Big[\mathrm{Ham}\big(\hat{\boldsymbol{a}}_T, \mathrm{sign}(\boldsymbol{\theta})\big)\Big],$$

where

$$\mathrm{Ham}(\boldsymbol{u}, \boldsymbol{v}) \triangleq \sum_{j=1}^d \mathbb{1}\{u_j \neq v_j\}.$$

Let $S = \sum_{t=1}^T \sigma_t^{-2}$ denote the total precision. Fix $j \in [d]$ and consider neighboring parameters that differ only on coordinate $j$:

$$\boldsymbol{\theta}_j^{(j,+)} = +c, \quad \boldsymbol{\theta}_j^{(j,-)} = -c, \qquad \boldsymbol{\theta}_k^{(j,+)} = \boldsymbol{\theta}_k^{(j,-)} \in \{\pm c\} \ (k \neq j).$$

Let $\mathbb{P}_+^{(j)} \equiv \mathbb{P}_{\boldsymbol{\theta}^{(j,+)}}$ and $\mathbb{P}_-^{(j)} \equiv \mathbb{P}_{\boldsymbol{\theta}^{(j,-)}}$ be the laws of the full transcript under these two instances. By the chain rule for KL and the fact that $\boldsymbol{a}_t = \pi_t(H_{t-1})$ contributes to KL,

$$\mathrm{KL}(\mathbb{P}_+^{(j)}\|\mathbb{P}_-^{(j)}) = \sum_{t=1}^T \mathbb{E}\big[\mathrm{KL}\big(\mathcal{N}(\mu_t^{(j,+)}, \sigma_t^2)\|\mathcal{N}(\mu_t^{(j,-)}, \sigma_t^2)\big)\big],$$

where $\mu_t^{(j,\pm)} = \langle \boldsymbol{\theta}^{(j,\pm)}, \boldsymbol{a}_t\rangle$. The means differ only on coordinate $j$, so $\mu_t^{(j,+)} - \mu_t^{(j,-)} = (+c - (-c))a_{t,j} = 2c\, a_{t,j}$ and thus

$$\mathrm{KL}\big(\mathcal{N}(\mu_t^{(j,+)}, \sigma_t^2)\,\|\,\mathcal{N}(\mu_t^{(j,-)}, \sigma_t^2)\big) = \frac{(\mu_t^{(j,+)} - \mu_t^{(j,-)})^2}{2\sigma_t^2} = \frac{(2c\, a_{t,j})^2}{2\sigma_t^2} = \frac{2c^2}{\sigma_t^2},$$

since $a_{t,j}^2 = 1$. Summing over $t$ yields the instance divergence

$$\mathrm{KL}(\mathbb{P}_+^{(j)} \| \mathbb{P}_-^{(j)}) = 2c^2 \sum_{t=1}^{T} \frac{1}{\sigma_t^2} = 2c^2 S.$$

**Step 2 (Le Cam + Pinsker $\Rightarrow$ per-coordinate average error).** Apply Lemma C.1 with the test $\phi = \mathbb{1}\{\hat{s}_j = +1\}$ between $P_+^{(j)}$ and $P_-^{(j)}$, and bound the TV distance by Lemma C.2. We get

$$\frac{1}{2}\big(P_+^{(j)}\{\hat{s}_j \neq +1\} + P_-^{(j)}\{\hat{s}_j \neq -1\}\big) \geq \frac{1}{2}\Big(1 - \sqrt{\tfrac{1}{2}\mathrm{KL}(P_+^{(j)} \| P_-^{(j)})}\Big) = \frac{1}{2}\big(1 - c\sqrt{S}\big).$$

Choose $c = \frac{1}{4}S^{-1/2}$ to obtain the uniform per-coordinate bound

$$\frac{1}{2}\big(P_+^{(j)}\{\hat{s}_j \neq +1\} + P_-^{(j)}\{\hat{s}_j \neq -1\}\big) \geq \frac{3}{8} \qquad \text{for all } j \in [d].$$

**Step 3 (aggregate to $d$ coordinates under Hamming loss).** This part requires the following lemma. The proof of this Lemma is defer to Appendix C.1.

**Lemma C.4** (From two-point bounds to a $d$-dimensional Hamming-risk lower bound)**.** Let $\Theta = \{\pm c\}^d$ and put the uniform prior on $\Theta$. Let $H_T$ denote the full transcript and let $\hat{s} = \hat{s}(H_T) \in \{\pm 1\}^d$ be any estimator of $\mathrm{sign}(\boldsymbol{\theta})$. For each coordinate $j \in [d]$, fix two instances $\boldsymbol{\theta}^{(j,+)}, \boldsymbol{\theta}^{(j,-)} \in \Theta$ that differ only in coordinate $j$ (i.e., $\boldsymbol{\theta}_j^{(j,+)} = +c$, $\boldsymbol{\theta}_j^{(j,-)} = -c$ and $\boldsymbol{\theta}_k^{(j,+)} = \boldsymbol{\theta}_k^{(j,-)}$ for all $k \neq j$). Denote by $P_+^{(j)}$ and $P_-^{(j)}$ the corresponding laws of $H_T$ under these two instances. Assume that for every $j$,

$$\frac{1}{2}\Big(P_+^{(j)}\{\hat{s}_j \neq +1\} + P_-^{(j)}\{\hat{s}_j \neq -1\}\Big) \geq \eta, \quad \text{for some } \eta \in (0, 1/2]. \tag{C.1}$$

Then the Bayes Hamming risk under the uniform prior satisfies

$$\mathbb{E}_{\boldsymbol{\theta}}\, \mathbb{E}_{\boldsymbol{\theta}}\big[\mathrm{Ham}(\hat{s}, \mathrm{sign}(\boldsymbol{\theta}))\big] \geq \eta\, d, \tag{C.2}$$

and, consequently, by Yao's minimax principle,

$$\inf_{\pi} \sup_{\boldsymbol{\theta} \in \Theta} \mathbb{E}_{\boldsymbol{\theta}}\big[\mathrm{Ham}(\hat{s}, \mathrm{sign}(\boldsymbol{\theta}))\big] \geq \eta\, d. \tag{C.3}$$

Invoke Lemma C.4 with $\eta = \frac{3}{8}$ to conclude that the Bayes Hamming risk under the uniform prior on $\Theta$ satisfies

$$\mathbb{E}_{\boldsymbol{\theta}}\, \mathbb{E}_{\boldsymbol{\theta}}\big[\mathrm{Ham}(\hat{s}, \mathrm{sign}(\boldsymbol{\theta}))\big] \geq \frac{3}{8}\, d.$$

By Lemma C.3 (Yao's principle), this also lower-bounds the minimax Hamming risk:

$$\inf_{\pi} \sup_{\boldsymbol{\theta} \in \Theta} \mathbb{E}_{\boldsymbol{\theta}}\big[\mathrm{Ham}(\hat{s}, \mathrm{sign}(\boldsymbol{\theta}))\big] \geq \frac{3}{8}\, d.$$

**Step 4 (convert Hamming loss to simple regret).** In $\Theta = \{\pm c\}^d$, each coordinate mistake costs exactly $2c$ in value. Therefore

$$\inf_{\pi} \sup_{\boldsymbol{\theta} \in \Theta} \mathbb{E}_{\boldsymbol{\theta}}\big[\mathrm{SR}(\pi, \boldsymbol{\theta}, T)\big] \geq 2c \cdot \frac{3}{8}\, d = \frac{3}{4}\, c\, d = \frac{3}{16}\, d\, S^{-1/2},$$

where we used $c = \frac{1}{4}S^{-1/2}$. $\qquad\square$

## C.1 PROOF OF LEMMA C.4

*Proof of Lemma C.4.* By definition of Hamming distance,

$$\mathrm{Ham}(\hat{s}, \mathrm{sign}(\boldsymbol{\theta})) = \sum_{j=1}^{d} \mathbb{1}\{\hat{s}_j \neq \mathrm{sign}(\boldsymbol{\theta}_j)\}.$$

Taking expectation under the model instance $\boldsymbol{\theta}$ and then averaging over the uniform prior on $\Theta$, the Bayes Hamming risk equals

$$\mathbb{E}_{\boldsymbol{\theta}}\,\mathbb{E}_{\boldsymbol{\theta}}\big[\mathrm{Ham}(\hat{\boldsymbol{s}}, \mathrm{sign}(\boldsymbol{\theta}))\big] = \sum_{j=1}^{d} \mathbb{E}_{\boldsymbol{\theta}}\,\mathbb{P}_{\boldsymbol{\theta}}\big\{\hat{s}_j \neq \mathrm{sign}(\boldsymbol{\theta}_j)\big\}. \tag{C.4}$$

Fix a coordinate $j \in [d]$. Write $\boldsymbol{\theta} = (\theta_j, \boldsymbol{\theta}_{-j})$, and condition on $\boldsymbol{\theta}_{-j}$. Under the *uniform* prior on $\Theta$, we have $\mathbb{P}\{\theta_j = +c \mid \boldsymbol{\theta}_{-j}\} = \mathbb{P}\{\theta_j = -c \mid \boldsymbol{\theta}_{-j}\} = \frac{1}{2}$. Hence, conditionally on $\boldsymbol{\theta}_{-j}$, the law of the transcript $H_T$ is the equal mixture

$$\mathcal{M}_{\boldsymbol{\theta}_{-j}}^{(j)} = \tfrac{1}{2}\,P_+^{(j)} + \tfrac{1}{2}\,P_-^{(j)}, \tag{C.5}$$

where $P_+^{(j)}$ and $P_-^{(j)}$ are the endpoint laws that differ only in coordinate $j$ (with $\boldsymbol{\theta}_{-j}$ held fixed).

**Identify the conditional Bayes error for coordinate $j$ with the average two-point error.** Consider the indicator loss for estimating $\mathrm{sign}(\theta_j)$ by the (measurable) decision rule $\hat{s}_j(H_T) \in \{\pm 1\}$. The conditional Bayes error probability for coordinate $j$, given $\boldsymbol{\theta}_{-j}$, is

$$\mathrm{Err}_j(\boldsymbol{\theta}_{-j}) \triangleq \mathbb{E}_{H_T \sim \mathcal{M}_{\boldsymbol{\theta}_{-j}}^{(j)}}\Big[ \tfrac{1}{2}\,\mathbb{1}\{\hat{s}_j(H_T) \neq +1\} + \tfrac{1}{2}\,\mathbb{1}\{\hat{s}_j(H_T) \neq -1\}\Big].$$

Using (C.5), this equals the *average* of the two endpoint errors:

$$\mathrm{Err}_j(\boldsymbol{\theta}_{-j}) = \tfrac{1}{2}\,P_+^{(j)}\{\hat{s}_j \neq +1\} + \tfrac{1}{2}\,P_-^{(j)}\{\hat{s}_j \neq -1\}. \tag{C.6}$$

By the assumption (C.1), we have, for every $\boldsymbol{\theta}_{-j}$,

$$\mathrm{Err}_j(\boldsymbol{\theta}_{-j}) \geq \eta. \tag{C.7}$$

**Average over $\theta_{-j}$ and sum over $j$.** By the tower property (law of total expectation),

$$\mathbb{E}_{\boldsymbol{\theta}}\,\mathbb{P}_{\boldsymbol{\theta}}\big\{\hat{s}_j \neq \mathrm{sign}(\boldsymbol{\theta}_j)\big\} = \mathbb{E}_{\boldsymbol{\theta}_{-j}}\Big[\,\mathrm{Err}_j(\boldsymbol{\theta}_{-j})\,\Big].$$

Combining with (C.7) yields

$$\mathbb{E}_{\boldsymbol{\theta}}\,\mathbb{P}_{\boldsymbol{\theta}}\big\{\hat{s}_j \neq \mathrm{sign}(\boldsymbol{\theta}_j)\big\} \geq \eta \qquad \text{for every } j \in [d]. \tag{C.8}$$

Summing (C.8) over $j = 1, \dots, d$ and using (C.4) gives

$$\mathbb{E}_{\boldsymbol{\theta}}\,\mathbb{E}_{\boldsymbol{\theta}}\big[\mathrm{Ham}(\hat{\boldsymbol{s}}, \mathrm{sign}(\boldsymbol{\theta}))\big] = \sum_{j=1}^{d} \mathbb{E}_{\boldsymbol{\theta}}\,\mathbb{P}_{\boldsymbol{\theta}}\big\{\hat{s}_j \neq \mathrm{sign}(\boldsymbol{\theta}_j)\big\} \geq \eta\,d,$$

which is (C.2).

**From Bayes to Minimax.** By Yao's minimax principle (Lemma C.3), the Bayes risk under the uniform prior lower-bounds the minimax (worst-case) risk over $\Theta$ of any deterministic policy:

$$\inf_{\pi}\,\sup_{\boldsymbol{\theta} \in \Theta}\,\mathbb{E}_{\boldsymbol{\theta}}\big[\mathrm{Ham}(\hat{\boldsymbol{s}}, \mathrm{sign}(\boldsymbol{\theta}))\big] \geq \mathbb{E}_{\boldsymbol{\theta}}\,\mathbb{E}_{\boldsymbol{\theta}}\big[\mathrm{Ham}(\hat{\boldsymbol{s}}, \mathrm{sign}(\boldsymbol{\theta}))\big] \geq \eta\,d,$$

which is (C.3). This completes the proof. $\qquad\square$

## D  PROOF OF THEOREM E.3

**Lemma D.1.** If we follow Algorithm 2 to choose the action $\mathbf{a}_{1:T}$, and compute $\hat{\boldsymbol{\theta}}_T$ and $V_T$, then it holds that

$$V_T \succeq \Big[\sum_{t=1}^{T} \frac{1}{\sigma_t^2} - \sum_{i=1}^{|\mathrm{supp}(\pi)|} \frac{1}{[\sigma_T^{(i)}]^2}\Big] V(\pi).$$

*Proof.* Consider the action $\mathbf{a}_m \in \mathcal{A}$ such that $\pi(\mathbf{a}_m) > 0$ and $\sum_{\tau \in \mathcal{T}_T(\mathbf{a})} \frac{1}{\sigma_\tau^2 \cdot \pi(\mathbf{a})}$ is minimized. It is straightforward to see that

$$V_T \succeq \sum_{\mathbf{a} \in \mathcal{A}} \left[ \sum_{\tau \in \mathcal{T}_T(\mathbf{a}_m)} \frac{1}{\sigma_\tau^2 \cdot \pi(\mathbf{a}_m)} \right] \pi(\mathbf{a}) \mathbf{a} \mathbf{a}^\top = \sum_{\tau \in \mathcal{T}_T(\mathbf{a}_m)} \frac{1}{\sigma_\tau^2 \cdot \pi(\mathbf{a}_m)} V(\pi). \tag{D.1}$$

Then it suffices to lower bound $\sum_{\tau \in \mathcal{T}_T(\mathbf{a}_m)} \frac{1}{\sigma_\tau^2 \cdot \pi(\mathbf{a}_m)}$.

We consider the arms in $\mathrm{supp}(\pi) \backslash \{\mathbf{a}_m\}$. For the round $t(\mathbf{a})$ when the arm $\mathbf{a}$ is pulled and $\sum_{\tau \in \mathcal{T}_t(\mathbf{a})} \frac{1}{\sigma_\tau^2 \cdot \pi(\mathbf{a})} \geq \sum_{\tau \in \mathcal{T}_T(\mathbf{a}_m)} \frac{1}{\sigma_\tau^2 \cdot \pi(\mathbf{a}_m)}$, it is guaranteed by the selection rule that $\mathbf{a}$ will not be pulled in the subsequent rounds. We denote by $\sigma(\mathbf{a})$ the last variance observed when pulling the arm $\mathbf{a}$. Then we have

$$\sum_{\mathbf{a} \in \mathrm{supp}(\pi) \backslash \{\mathbf{a}_m\}} \pi(\mathbf{a}) \sum_{\tau \in \mathcal{T}_T(\mathbf{a})} \frac{1}{\sigma_\tau^2 \cdot \pi(\mathbf{a})} \leq \sum_{\mathbf{a} \in \mathrm{supp}(\pi) \backslash \{\mathbf{a}_m\}} \pi(\mathbf{a}) \left[ \sum_{\tau \in \mathcal{T}_T(\mathbf{a}_m)} \frac{1}{\sigma_\tau^2 \cdot \pi(\mathbf{a}_m)} + \frac{1}{[\sigma(\mathbf{a})]^2 \cdot \pi(\mathbf{a})} \right]$$

$$\leq [1 - \pi(\mathbf{a}_m)] \sum_{\tau \in \mathcal{T}_T(\mathbf{a}_m)} \frac{1}{\sigma_\tau^2 \cdot \pi(\mathbf{a}_m)} + \sum_{\mathbf{a} \in \mathrm{supp}(\pi) \backslash \{\mathbf{a}_m\}} \frac{1}{[\sigma(\mathbf{a})]^2}$$

$$\leq [1 - \pi(\mathbf{a}_m)] \sum_{\tau \in \mathcal{T}_T(\mathbf{a}_m)} \frac{1}{\sigma_\tau^2 \cdot \pi(\mathbf{a}_m)} + \sum_{i=1}^{|\mathrm{supp}(\pi)|-1} \frac{1}{[\sigma_T^{(i)}]^2}, \tag{D.2}$$

where the first inequality is due to the definition of $\sigma(\mathbf{a})$ and the last inequality follows from the fact that $\{\sigma(\mathbf{a})\}_{\mathbf{a} \in \mathrm{supp}(\pi) \backslash \{\mathbf{a}_m\}}$ are the variance signals observed at distinct rounds.

Rearranging (D.2), we obtain that

$$\sum_{\mathbf{a} \in \mathrm{supp}(\pi)} \pi(\mathbf{a}) \sum_{\tau \in \mathcal{T}_T(\mathbf{a})} \frac{1}{\sigma_\tau^2 \cdot \pi(\mathbf{a})} \leq \sum_{\tau \in \mathcal{T}_T(\mathbf{a}_m)} \frac{1}{\sigma_\tau^2 \cdot \pi(\mathbf{a}_m)} + \sum_{i=1}^{|\mathrm{supp}(\pi)|-1} \frac{1}{[\sigma_T^{(i)}]^2}. \tag{D.3}$$

Note that LHS of (D.3) is exactly $\sum_{t=1}^T \frac{1}{\sigma_t^2}$, which further indicates that

$$\sum_{\tau \in \mathcal{T}_T(\mathbf{a}_m)} \frac{1}{\sigma_\tau^2 \cdot \pi(\mathbf{a}_m)} \geq \sum_{t=1}^T \frac{1}{\sigma_t^2} - \sum_{i=1}^{|\mathrm{supp}(\pi)|-1} \frac{1}{[\sigma_T^{(i)}]^2}.$$

As a result, we have

$$V_T \succeq \left[ \sum_{t=1}^T \frac{1}{\sigma_t^2} - \sum_{i=1}^{|\mathrm{supp}(\pi)|-1} \frac{1}{[\sigma_T^{(i)}]^2} \right] V(\pi)$$

following (D.1). $\qquad \square$

**Lemma D.2.** In Algorithm 2, for any arm $\mathbf{a} \in \mathcal{A}$, with probability at least $1 - \delta$, we have

$$|\langle \hat{\boldsymbol{\theta}}_T - \boldsymbol{\theta}^*, \mathbf{a} \rangle| \leq \|\mathbf{a}\|_{V_T^{-1}} \sqrt{2 \log(2|\mathcal{A}|/\delta)}$$

$$\leq \|\mathbf{a}\|_{V(\pi)^{-1}} \sqrt{2 \log(2|\mathcal{A}|/\delta) / \left[ \sum_{t=1}^T \frac{1}{\sigma_t^2} - \sum_{i=1}^{|\mathrm{supp}(\pi)|} \frac{1}{[\sigma_T^{(i)}]^2} \right]}.$$

*Proof.* From the definition of $\hat{\boldsymbol{\theta}}_T$ and $V_T$, it is straightforward to obtain the following inequality:

$$|\langle \hat{\boldsymbol{\theta}}_T - \boldsymbol{\theta}^*, \mathbf{a} \rangle| = |\langle V_T^{-1} \sum_{t=1}^T \sigma_t^{-2} r_t \mathbf{a}_t - V_T^{-1} V_T \cdot \boldsymbol{\theta}^*, \mathbf{a} \rangle|$$

$$= |\langle V_T^{-1} \sum_{t=1}^T \sigma_t^{-2} \mathbf{a}_t \eta_t, \mathbf{a} \rangle|$$

$$= \Big| \sum_{t=1}^{T} \sigma_t^{-2} \eta_t \langle \mathbf{a}_t, V_T^{-1} \mathbf{a} \rangle \Big|. \tag{D.4}$$

Applying Lemma D.3 to (D.4), we have that with probability at least $1 - \delta$,

$$|\langle \hat{\boldsymbol{\theta}}_T - \boldsymbol{\theta}^*, \mathbf{a} \rangle| \leq \sqrt{2 \sum_{t=1}^{T} \sigma_t^{-2} \langle \mathbf{a}_t, V_T^{-1} \mathbf{a} \rangle^2 \log(2/\delta)}$$

$$\leq \sqrt{2 \sum_{t=1}^{T} [\sigma_t^{-1} \mathbf{a}_t^\top V_T^{-1} \mathbf{a}]^2 \log(2/\delta)}$$

$$= \sqrt{2 \sum_{t=1}^{T} \mathbf{a}^\top V_T^{-1} \sigma_t^{-1} \mathbf{a}_t \sigma_t^{-1} \mathbf{a}_t^\top V_T^{-1} \mathbf{a} \log(2/\delta)}$$

$$\leq \sqrt{2 \mathbf{a}^\top V_T^{-1} \mathbf{a} \log(2/\delta)},$$

where the last inequality is due to the fact that $\sum_{t=1}^{T} \sigma_t^{-2} \mathbf{a}_t \mathbf{a}_t^\top = V_T$.

By Lemma D.1, we further have

$$|\langle \hat{\boldsymbol{\theta}}_T - \boldsymbol{\theta}^*, \mathbf{a} \rangle| \leq \|\mathbf{a}\|_{V_T^{-1}} \sqrt{2 \log(2/\delta)}$$

$$\leq \|\mathbf{a}\|_{V(\pi)^{-1}} \sqrt{2 \log(2/\delta) / \Big[ \sum_{t=1}^{T} \frac{1}{\sigma_t^2} - \sum_{i=1}^{|\mathrm{supp}(\pi)|} \frac{1}{[\sigma_T^{(i)}]^2} \Big]},$$

from which the desired result follows by taking a union bound over all $\mathbf{a} \in \mathcal{A}$. $\qquad \square$

**Lemma D.3** (Hoeffding's inequality). *Let $\{x_i\}_{i=1}^n$ be a stochastic process, $\{\mathcal{G}_i\}_i$ be a filtration so that for all $i \in [n]$, $x_i$ is $\mathcal{G}_i$-measurable, while $\mathbb{E}[x_i | \mathcal{G}_{i-1}] = 0$ and $x_i | \mathcal{G}_{i-1}$ is a $\sigma_i$-sub-Gaussian random variable. Then, for any $t > 0$, with probability at least $1 - \delta$, it holds that*

$$\sum_{i=1}^{n} x_i \leq \sqrt{2 \sum_{i=1}^{n} \sigma_i^2 \log(1/\delta)}.$$

**Theorem D.4** (Simple Regret of Algorithm 2, restatement of Theorem 5.3). *Suppose that $\mathcal{A} \subset \mathbb{R}^d$ is compact and $\mathrm{span}(\mathcal{A}) = \mathbb{R}^d$. If we follow Algorithm 2, then it holds that with probability at least $1 - \delta$,*

$$\langle \boldsymbol{\theta}^*, \mathbf{a}^* \rangle - \langle \boldsymbol{\theta}^*, \mathbf{a}_{T+1} \rangle \leq 2 \sqrt{d \log(|\mathcal{A}|/\delta) / \Big[ \sum_{t=1}^{T} \frac{1}{\sigma_t^2} - \sum_{i=1}^{4d \log \log d + 16} \frac{1}{[\sigma_T^{(i)}]^2} \Big]}.$$

*Proof.* From the definition of $\mathbf{a}_{T+1}$, we have

$$\langle \boldsymbol{\theta}^*, \mathbf{a}^* \rangle - \langle \boldsymbol{\theta}^*, \mathbf{a}_{T+1} \rangle = \langle \hat{\boldsymbol{\theta}}_T - \boldsymbol{\theta}^*, \mathbf{a}_{T+1} \rangle + \langle \boldsymbol{\theta}^* - \hat{\boldsymbol{\theta}}_T, \mathbf{a}^* \rangle + \langle \hat{\boldsymbol{\theta}}_T, \mathbf{a}^* - \mathbf{a}_{T+1} \rangle$$

$$\leq |\langle \hat{\boldsymbol{\theta}}_T - \boldsymbol{\theta}^*, \mathbf{a}_{T+1} \rangle| + |\langle \hat{\boldsymbol{\theta}}_T - \boldsymbol{\theta}^*, \mathbf{a}^* \rangle|$$

$$\leq (\|\mathbf{a}_{T+1}\|_{V(\pi)^{-1}} + \|\mathbf{a}^*\|_{V(\pi)^{-1}}) \sqrt{2 \log(2|\mathcal{A}|/\delta) / \Big[ \sum_{t=1}^{T} \frac{1}{\sigma_t^2} - \sum_{i=1}^{|\mathrm{supp}(\pi)|} \frac{1}{[\sigma_T^{(i)}]^2} \Big]}$$

$$\leq 2 \sqrt{d \log(2|\mathcal{A}|/\delta) / \Big[ \sum_{t=1}^{T} \frac{1}{\sigma_t^2} - \sum_{i=1}^{4d \log \log d + 16} \frac{1}{[\sigma_T^{(i)}]^2} \Big]},$$

where the first inequality follows from the definition of $\mathbf{a}_{T+1}$, the second inequality is due to Lemma D.2 and the last inequality is due to Theorem 5.2. $\qquad \square$

---

**Algorithm 3** Variance-Aware Exploration with Elimination (`VAEE` for heavy-tailed noise)

---

**Require:** $\mathcal{A} \subset \mathbb{R}^d, \delta$.

1: Initialize $V_0 \leftarrow \lambda I_d, \hat{\boldsymbol{\theta}}_0 \leftarrow 0, \mathcal{A}_1 \leftarrow \mathcal{A}$.
2: **for** $t = 1, \ldots, T$ **do**
3:      Pull the action $\mathbf{a}_t \leftarrow \max_{\mathbf{e} \in \mathcal{A}_t} \|\mathbf{e}\|_{V_{t-1}^{-1}}$.
4:      The agent receives the reward $r_t$ and the variance $\sigma_t$.
5:      Calculate $V_t \leftarrow V_{t-1} + \sigma_t^{-2} \mathbf{a}_t \mathbf{a}_t^\top$.
6:      Calculate

$$\hat{\boldsymbol{\theta}}_t \leftarrow \underset{\|\boldsymbol{\theta}\|_2 \leq 1}{\operatorname{argmin}} \frac{\lambda}{2} \|\boldsymbol{\theta}\|_2^2 + \sum_{i=1}^t \ell_{\tau_i}\left(\frac{r_i - \langle \mathbf{a}_i, \boldsymbol{\theta} \rangle}{\sigma_i}\right),$$

     where $\tau_i = \tau_0 \cdot \dfrac{\sqrt{1 + \sigma_i^{-2} \|\mathbf{a}_i\|_{V_{i-1}^{-1}}^2}}{\sigma_i^{-1} \|\mathbf{a}_i\|_{V_{i-1}^{-1}}}$.
7:      Set confidence set as follows

$$\mathcal{C}_t \leftarrow \{\boldsymbol{\theta} \mid \|\boldsymbol{\theta} - \hat{\boldsymbol{\theta}}_t\|_{V_t^{-1}}^2 \leq \beta_t\}.$$

8:      Eliminate low rewarding arms: $\mathcal{A}_{t+1} \leftarrow \{\mathbf{a} \in \mathcal{A}_t : \max_{\mathbf{e} \in A_t} \min_{\boldsymbol{\theta} \in \mathcal{C}_t} \langle \boldsymbol{\theta}, \mathbf{e} \rangle \leq \max_{\boldsymbol{\theta} \in \mathcal{C}_t} \langle \boldsymbol{\theta}, \mathbf{a} \rangle\}$.
9: **end for**

---

## E    EXTENSION TO HEAVY-TAILED NOISE

In this section, we extend our results to the setting where the noise is heavy-tailed. Specifically, we consider the following assumption on the noise.

**Assumption E.1.** For any round $t$ ($t \geq 1$), the noise $\eta_t$ satisfies that

$$\mathbb{E}[\eta_t | \mathbf{a}_{1:t}, \eta_{1:t-1}] = 0, \quad \mathbb{E}[\eta_t^2 | \mathbf{a}_{1:t}, \eta_{1:t-1}] \leq \sigma_t^2.$$

This assumption is more general than the sub-Gaussian assumption on the noise, which only requires the second moment of the noise to be bounded.

To handle the heavy-tailed noise, we consider the following adaptive pseudo-Huber regression estimator (Ruppert, 2004; Sun, 2021; Li & Sun, 2024):

$$\ell_\tau(x) := \tau \cdot \left(\sqrt{\tau^2 + x^2} - \tau\right), \tag{E.1}$$

where $\tau > 0$ is a robustification parameter. The pseudo-Huber loss behaves like the squared loss when $|x|$ is small, and behaves like the absolute loss when $|x|$ is large, which is firstly applied by Li & Sun (2024) into the heteroscedastic linear bandit setting.

**Lemma E.2** (Theorem 2.1, Li & Sun 2024). Let $\kappa = d \cdot \log(1 + T/d\sigma_{\min}^2)$. If we set $\tau_0 \geq \max\{\sqrt{2\kappa}, 2\sqrt{d}\}/\sqrt{\log(2T^2/\delta)}$, then with probability at least $1 - 4\delta$, it holds for all $t \in [T]$ that

$$\|\hat{\boldsymbol{\theta}}_t - \boldsymbol{\theta}^*\|_{V_t} \leq \beta_t := 32\left(\frac{\kappa}{\tau_0} + \sqrt{\kappa \log \frac{2t^2}{\delta}} + \tau_0 \log \frac{2t^2}{\delta}\right) + 5\sqrt{\lambda}.$$

**Theorem E.3** (Simple Regret of Algorithm 3). Consider the linear bandit problem with heavy-tailed noise satisfying Assumption E.1. If we set $\tau_0 = \Theta(\sqrt{d})$ and $\lambda = 1$ in Algorithm 3, then with probability at least $1 - 4\delta$, it holds that

$$\mathrm{SR}(T) = \tilde{O}(\sqrt{d}) \cdot \min_{1 \leq k \leq T+1} \left\{ x = \sqrt{\frac{\iota(T) - k + 1}{\sum_{i=k}^T \frac{1}{[\sigma_T^{(i)}]^2}}} \,\middle|\, x \in [\sigma_T^{(k-1)}, \sigma_T^{(k)}] \right\},$$

where $\iota(T) = 2d \log\left(\frac{d + \sum_{\tau \in [T]} \sigma_\tau^{-2}}{d}\right)$. Recall that $\{\sigma_T^{(i)}\}_{i=1}^T$ is the sorted sequence of $\{\sigma_t\}_{t=1}^T$ in the ascending order.

*Proof.* The proof follows the same line as the proof of Theorem 4.1, with the only difference being the confidence radius $\beta_t$. By setting $\tau_0 = \Theta(\sqrt{d})$, we have $\beta_t = \tilde{O}(\sqrt{d})$. Following the same analysis as in Theorem 4.1, we can obtain the desired result. $\qquad\square$

