# OpenReview forum: "Breaking the Total Variance Barrier: Sharp Sample Complexity for Linear Heteroscedastic Bandits with Fixed Action Set"
_ICLR.cc/2026/Conference — ICLR 2026 Poster_

### Official Review · Reviewer_btFg · 2025-10-26

**Soundness:** 3
**Presentation:** 3
**Contribution:** 3
**Rating:** 6
**Confidence:** 3

**Summary:**

The paper studies linear bandits with heteroscedastic noise under a fixed action set, aiming to improve simple regret beyond the classical dependence on the cumulative noise variance $\Lambda$ (“total variance”). It proposes two variance-adaptive algorithms:

* VAEE (Variance-Aware Exploration with Elimination) for large/continuous action sets, which selects actions by maximizing information gain while progressively eliminating arms.
* VAGD (Variance-Adaptive G-Optimal Design) for finite action sets, combining approximate G-optimal design with variance-adaptive sampling.

Both algorithms yield harmonic-mean–type simple-regret bounds of the form
*  $\tilde{O}\left(d \Big[\sum_t 1/\sigma_t^2 - \sum_{i=1}^{\tilde O(d)} 1/\sigma_{(i)}^2\Big]^{-1/2}\right)$  for VAEE and

* $\tilde{O}\left(\sqrt{d\log |{\mathcal A}|}  \Big[\sum_t 1/\sigma_t^2 - \sum_{i=1}^{\tilde O(d)} 1/\sigma_{(i)}^2\Big]^{-1/2}\right)$ for VAGD,

which the authors position as breaking the classic $\sqrt{\Lambda}$ barrier in simple regret. A matching instance-dependent lower bound (Theorem 6.1) shows a $\Omega \big(d (\sum_t 1/\sigma_t^2)^{-1/2}\big)$ dependence.

**Strengths:**

1. **Clear statement of contributions and rates (with concrete formulas)**.
The abstract precisely states the harmonic-mean–dependent rates (including the subtracting of the  $\tilde O(d)$ smallest variances) for both VAEE and VAGD, and emphasizes that this breaks the $\sqrt{\Lambda}$ barrier in simple regret.

2. **Well-motivated setting and assumptions.**
The fixed action set requirement is explicitly justified: when contexts change adversarially, $\sqrt{\Lambda}$ remains unavoidable.

3. **Accessible algorithmic presentation.** Algorithm 1 (VAEE) and Algorithm 2 (VAGD) are spelled out line-by-line, including the elimination step for VAEE (Line 8) and the variance-weighted design logic for VAGD (Lines 1–8 and estimator definition).

4. **Thoughtful comparisons and intuition.** The case study in Section 4.1 contrasts VAEE with Weighted OFUL under a low-variance window and shows how VAEE reallocates exploration to weak coordinates; the discussion quantifies the differing simple-regret decay. Table 2 further compares simple and cumulative regrets across noise profiles.

5. **Lower bound with standard, transparent techniques.** Theorem 6.1’s proof uses Le Cam’s method and Pinsker’s inequality, clearly documented in Appendix C. This makes the paper more complete.

**Weaknesses:**

1. **Lack of clear separation between settings.**
   The paper studies both infinite and fixed action sets, but Table 1 does not indicate which results correspond to each case. A new column or clearer labeling would make the distinction explicit.

2. **Unsubstantiated claim of “breaking the $\sqrt{\Lambda}$ barrier.”**
   The main text claims improvement over the $\sqrt{\Lambda}$ bound but does not point to the derivation showing this. Equation (4.2) and Appendix B contain the critical argument and should be cited when stating the main contributions.

3. **Unclear source of improvement in Algorithms 1–2.**
   It remains ambiguous whether the tighter regret stems from (a) the elimination step, (b) a sharper analysis, or (c) the fixed-action assumption. A short discussion quantifying computational or analytical contributions would improve transparency.

4. **Missing or delayed notation definitions.**
   Several symbols appear before definition. For example,
     - the global window $W$ (in line **278**),
     - the terms $\mathrm{UCB}_x(t)$, $\mathrm{UCB}_0(t)$, $\mathrm{UCB}_1(t)$ (in lines **284–286**), and
     - $\mathcal{P}(\mathcal{A}*{\ell})$ (in line 400).

   Early or inline definitions would improve readability.

5. **Lack of Empirical Illustration**

    Despite the clear theoretical development and the well-specified variance settings in Table 2 (e.g., fast-decay, spiky variance profiles), the paper provides no empirical or synthetic validation.
    Since the variance is explicitly given, a simple simulation would be straightforward and would help demonstrate the practical relevance of the proposed regret bounds and constants.
    The absence of such experiments weakens the overall empirical support for the claimed improvements.

**Questions:**

1. **Isolation of the Key Driver**

   For VAEE, what component is primarily responsible for achieving the harmonic-mean regret rate — (i) the elimination step (Algorithm 1, Line 8), (ii) the variance-weighted estimator and confidence sets, or (iii) the fixed-action structure noted in Remark 3.1?
   A brief ablation-style analysis or a theoretical “swap-in/swap-out” comparison would help clarify the main driver of improvement.

2. **Complexity–Regret Trade-offs**

   What are the computational complexities of VAEE (both per round and overall, including elimination checks) and VAGD (computing or approximating the design with $|\mathrm{supp}(\pi)| \le 4d\log\log d + 16$)?
   Some discussion on the scalability or possible computational bottlenecks would strengthen the practical perspective.

3. **Unknown or Misspecified $\sigma_t$**

   Assumption 3.2 assumes conditionally $\sigma_t$-sub-Gaussian noise with known bounds $(\sigma_{\min}, \sigma_{\max})$.
   How robust are the results if $\sigma_t$ is unknown or mis-specified? Additionally, can the heavy-tailed extension mentioned in the remarks achieve the same harmonic-mean dependence? A short discussion in the main text would help.

4. **Non-Spanning Action Sets**

   Theorem 5.2 assumes that $\mathcal{A}$ spans $\mathbb{R}^d$. What if this condition does not hold? Could the authors formalize the subspace reduction and restate Theorem 5.3 with $d'$ denoting the rank of $\mathcal{A}$? In other words, if $\mathcal{A}$ lies within a lower-dimensional linear subspace of dimension $d'$, can the existing bounds be adapted by replacing $d$ with $d'$?
   Clarifying this case would enhance the completeness and generality of the theoretical results.

---

> ### Author Response · Authors · 2025-11-21
> **Response to Reviewer btFg**
>
> Thank you for your positive feedback and insightful comments!
>
> ---
> **Q1.** The paper studies both infinite and fixed action sets, but Table 1 does not indicate which results correspond to each case. A new column or clearer labeling would make the distinction explicit.
>
> **A1.** We thank the reviewer for this helpful comment. To eliminate any ambiguity, we have revised Table 1 (on page 3) to clearly indicate, for each result, whether it pertains to the infinite-action or fixed-action setting.
>
> ---
>
> **Q2.** The main text claims improvement over the $\sqrt{\Lambda}$ bound but does not point to the derivation showing this. Equation (4.2) and Appendix B contain the critical argument and should be cited when stating the main contributions.
>
> **A2.** We thank the reviewer for this constructive suggestion. We agree that explicitly citing the technical derivation strengthens the presentation and makes the improvement over $\sqrt{\Lambda}$ more transparent to readers.
>
> In Section 1, we will add reference to (4.2) and Appendix B:
> > "To see why our harmonic-mean dependent rate is strictly better than the $\sqrt{\Lambda}$-type bounds, note that by the harmonic-mean/arithmetic-mean inequality, we have $T/(\sum_t 1/\sigma_t^2) \leq (\sum_t \sigma_t^2)/T = \Lambda/T$. This implies our simple regret bound scales as $\widetilde{O}(d\sqrt{T/\sum_t 1/\sigma_t^2}) \leq \widetilde{O}(d\sqrt{\Lambda/T^2})$, matching or improving upon prior bounds. **The formal derivation showing this relationship is presented in Equation (4.2) and detailed in Appendix B (see Remark B.5).**"
>
> We will also enhance Section 4.2 (Comparison with Weighted OFUL) by adding a forward reference:
>
> > "The improvement follows from the HM-AM inequality as detailed in Equation (4.2) and Remark B.5. We demonstrate..."
>
> Thank you for this suggestion, which will make our contribution more accessible and verifiable.
>
> ---
>
> **Q3.** It remains ambiguous whether the tighter regret stems from (a) the elimination step, (b) a sharper analysis, or \(c\) the fixed-action assumption. A short discussion quantifying computational or analytical contributions would improve transparency.
>
> **A3.** Thank you for raising this point. We clarify that breaking the $\sqrt{\Lambda}$ barrier requires **all three components working together**:
>
> (a) Fixed-action assumption.
>  Our lower bound (Theorem 6.1) shows that the harmonic-mean dependence is optimal under a fixed, finite action set, whereas for time-varying action sets the $\sqrt{\Lambda}$ rate is information-theoretically unavoidable \citep{he2025variance}. In this sense, the fixed-action assumption is a structural condition under which the improvement can be achieved.  We have added a brief discussion in Lines 70–73 to clarify this point.
>
> (b)Elimination + active exploration.
> The elimination step, combined with the active exploration rule, deliberately directs pulls toward informative directions in low-variance windows, while Weighted OFUL’s optimistic rule can keep selecting poorly informative arms. The two-dimensional example in Section~4.1 makes this gap explicit, showing that these new designs are genuinely necessary for our sharper variance-sequence-dependent guarantees.
>
> \(c\) Sharper, variance-sequence-aware analysis.
> The improvement in regret does not come from a minor tightening of existing proofs. Our main novelty is a variance-sequence-aware potential argument and the minimization structure in Theorem~4.1, which together track how information accumulates along the ordered variance sequence $\{\sigma_{(i)}\}$ and identify the optimal trade-off point over $k$. This framework is specific to fixed-action, heteroscedastic bandits and is not implied by existing variance-aware analyses.
>
> ---
>
> **Q4.** Missing or delayed notation definitions. Several symbols appear before definition.
>
> **A4.** We thank the reviewer for pointing this out.  We have also checked the  manuscript to ensure that any shorthand is introduced at its first appearance. These changes are purely expository and do not affect any results.

---

> ### Author Response · Authors · 2025-11-21
> **Response to Reviewer btFg Part II**
>
> ---
>
> **Q5.** What are the computational complexities of VAEE (both per round and overall, including elimination checks) and VAGD(computing or approximating the design with $|\text{supp}(\pi)| \le 4d \log\log d + 16$?
>
> **A5.** We thank the reviewer for this important question. We will add a dedicated subsection on computational complexity.
>
>
> We analyze the computational complexity of both proposed algorithms.
>
> #### VAEE (Algorithm 1)
>
> **Per-round operations:**
>
> 1. **Action selection (Line 3):** $a_t = \arg\max_{e \in \mathcal{A}\_t} \|e\|\_{V_{t-1}\^{-1}}$
>    - Requires computing $\|e\|\_{V\_{t-1}\^{-1}} = \sqrt{e\^\top V\_{t-1}\^{-1} e}$ for each $e \in \mathcal{A}_t$
>    - With $|\mathcal{A}_t| = n_t$ candidates: $O(n_t d^2)$ per candidate evaluation
>    - **Cost:** $O(|\mathcal{A}_t| \cdot d^2)$
>
> 2. **Matrix update (Line 5):** $V_t = V_{t-1} + \sigma_t^{-2} a_t a_t^\top$
>    - Rank-1 update: $O(d^2)$
>
> 3. **Parameter update (Line 6):** $\hat{\theta}\_t = V_t\^{-1} \sum_{s=1}\^t \sigma_s\^{-2} a_s r_s$
>    - Using Sherman-Morrison for incremental inverse: $O(d^2)$
>    - Alternatively, maintain $V_t^{-1}$ directly: $O(d^3)$
>
> 4. **Elimination step (Line 8):**
>    - For each $a \in \mathcal{A}\_t$: check if $\max\_{e \in \mathcal{A}\_t} \min\_{\theta \in \mathcal{C}\_t} \langle \theta, e \rangle \leq \max_{\theta \in \mathcal{C}\_t} \langle \theta, a \rangle$
>    - Computing $\max_{\theta \in \mathcal{C}\_t} \langle \theta, a \rangle = \langle \hat{\theta}\_t, a \rangle + \beta_t \|a\|\_{V_t\^{-1}}$: $O(d^2)$ per action
>    - **Cost:** $O(|\mathcal{A}_t| \cdot d^2)$
>
> **Per-round total:** $O(|\mathcal{A}_t| \cdot d^2)$
>
> **Evolution of $|\mathcal{A}_t|$:**
> - Initial: $|\mathcal{A}_1| = |\mathcal{A}|$ (potentially infinite)
> - After $\widetilde{O}(d)$ rounds: $|\mathcal{A}_t| = \widetilde{O}(d)$ with high probability (by elimination)
> - **Phase transition:** Rounds 1 to $\widetilde{O}(d)$ are expensive; subsequent rounds cost $O(d^3)$
>
> **Overall complexity:**
> - **Early phase** (first $\widetilde{O}(d)$ rounds): $O(|\mathcal{A}| \cdot d^3 \log T)$ if $\mathcal{A}$ is finite
> - **Late phase** (rounds $\widetilde{O}(d)$ to $T$): $O(T d^3)$
> - **Total:** $O(|\mathcal{A}| \cdot d^3 \log T + T d^3)$
>
> **For infinite action sets:** If $\mathcal{A}$ is a convex set (e.g., unit ball), Line 3 and Line 8 require solving convex optimization problems:
> - $\max_{e \in \mathcal{A}\_t} \|e\|\_{V\_{t-1}\^{-1}}^2 = \max_{e \in \mathcal{A}\_t} e^\top V_{t-1}\^{-1} e$ (convex quadratic program)
> - Using interior-point methods: $O(d^{3.5})$ per round (after elimination reduces $\mathcal{A}_t$ to implicit constraints)
>
> #### VAGD (Algorithm 2)
>
> **One-time preprocessing (Line 1):** Computing the G-optimal design $\pi$
>
> - **Exact computation:** Finding exact G-optimal design with $|\text{supp}(\pi)| \leq d(d+1)/2$ is expensive (semidefinite programming)
> - **Approximate design (Theorem 5.2):** Can compute $\pi$ with $|\text{supp}(\pi)| \leq 4d\log\log d + 16$ and $g(\pi) \leq 2d$
>
>   **Method (Lattimore et al., 2020):**
>   - Frank-Wolfe algorithm for D-optimal design
>   - **Iterations:** $O(d \log d \cdot \log(1/\epsilon))$ to achieve $g(\pi) \leq d(1 + \epsilon)$
>   - **Per iteration:** $O(|\mathcal{A}| d^2 + d^3)$ (solving $\arg\max_{a \in \mathcal{A}} \|a\|_{V(\pi_k)^{-1}}$ and matrix update)
>   - **Total preprocessing:** $O(|\mathcal{A}| d^3 \log d \log(1/\epsilon))$
>
> **Per-round operations (Lines 3-6):**
>
> 1. **Action selection (Line 4):** $a_t = \arg\min_{a \in \mathcal{A}} \frac{\sum_{\tau \in T(a)} 1/\sigma_\tau^2}{\pi(a)}$
>    - Only search over $\text{supp}(\pi)$ with $|\text{supp}(\pi)| = O(d \log\log d)$
>    - Maintain counters: $O(d \log\log d)$
>
> 2. **Matrix/parameter updates (Lines 5-6):** Same as VAEE
>    - $O(d^2)$ per round
>
> **Per-round total:** $O(d^2 + d\log\log d) = O(d^2)$
>
> **Overall complexity:**
> - **Preprocessing:** $O(|\mathcal{A}| d^3 \log d)$ (assuming $\epsilon = 1/\text{poly}(d)$)
> - **Online:** $O(T d^2)$
> - **Total:** $O(|\mathcal{A}| d^3 \log d + T d^2)$

---

> ### Author Response · Authors · 2025-11-21
> **Response to Reviewer btFg Part III**
>
> ---
>
> **Q6.** How robust are the results if $\sigma_t$ is unknown or mis-specified? Additionally, can the heavy-tailed extension mentioned in the remarks achieve the same harmonic-mean dependence? A short discussion in the main text would help.
>
> **A6.** As far as we know, the only framework which can handle heteroscedastic linear bandits with unknown variance is proposed by [2], which highly relies on their Theorem 2.
>
> > **Theorem 2.1 (Zhao et al., 2023a, paraphrased):** For variance-weighted martingale $\sum\_{t=1}\^T \hat{\sigma}\_t\^{-2} a\_t \eta\_t$ where $\hat{\sigma}\_t\^2$ are estimated variances, with high probability:
> > $$\left\|\sum\_{t=1}\^T \hat{\sigma}\_t\^{-2} a\_t \eta\_t\right\|_{\hat{V}\_T\^{-1}} \leq \widetilde{O}\left(\sqrt{d \sum\_{t=1}^T \sigma\_t\^2}\right) = \widetilde{O}(\sqrt{d\Lambda})$$
> > where $\hat{V}_T = \lambda I + \sum_t \hat{\sigma}_t^{-2} a_t a_t^\top$.
>
> With unknown variances, the $\sqrt{\Lambda}$-dependence appears **unavoidable** given current concentration techniques. And if variances are unknown in the heavy-tailed setting, we again face the $\sqrt{\Lambda}$ barrier due to reliance on concentration results analogous to Zhao et al. (2023a, Theorem 2.1). Breaking this barrier would require:
> - New concentration inequalities beyond Theorem 2.1 of Zhao et al. (2023a)
> - Or additional structural assumptions (e.g., variance predictability)
>
> We leave this open problem for future work.
>
> ---
>
> **Q7.** Theorem 5.2 assumes that $\mathcal{A}$ spans $\mathbb{R}^d$. What if this condition does not hold? Could the authors formalize the subspace reduction and restate Theorem 5.3 with $d'$ denoting the rank of $\mathcal{A}$? In other words, if $\mathcal{A}$ lies within a lower-dimensional linear subspace of dimension $d'$, can the existing bounds be adapted by replacing $d$ with $d'$? Clarifying this case would enhance the completeness and generality of the theoretical results.
>
>
> **A7.** This is an excellent observation that highlights an important but implicit aspect of the results. You are correct that when $\mathcal{A}$ does not span $\mathbb{R}^d$, a natural subspace reduction applies.
>
> **Setup:** Let $d' = \text{rank}(\mathcal{A})$ denote the dimension of $\text{span}(\mathcal{A})$, where $d' \leq d$. There exists an orthonormal basis $\{u_1, \ldots, u_{d'}\}$ for $\text{span}(\mathcal{A})$ and a matrix $U \in \mathbb{R}^{d \times d'}$ with orthonormal columns such that:
> - Every action $a \in \mathcal{A}$ can be written as $a = U\tilde{a}$ for some $\tilde{a} \in \mathbb{R}^{d'}$
> - The parameter $\theta^* \in \mathbb{R}^d$ can be projected as $\tilde{\theta}^* = U^\top \theta^*$
>
> **Key observation:** The reward satisfies
> $$ r\_t = \langle \theta\^*, a\_t \rangle + \eta\_t =  \langle U\^\top \theta\^\*, U^\top a\_t \rangle + \eta_t = \langle \tilde{\theta}\^\*, \tilde{a}_t \rangle + \eta_t$$
>
> This means the problem reduces to a $d'$-dimensional linear bandit on the action set $\tilde{\mathcal{A}} = \\{U^\top a : a \in \mathcal{A}\\}$.
>
> Thank you again for your suggestion. We will add this discussion into our revision.
>
> ---
> [1] Tor Lattimore & Csaba Szepesvári, Bandit Algorithms. Cambridge University Press, 2020
>
> [2] Zhao et al. Variance-dependent regret bounds for linear bandits and reinforcement learning: Adaptivity and computational efficiency. In *The Thirty Sixth Annual Conference on Learning Theory (COLT)*, pages 4977–5020. PMLR, 2023.

---

> > ### Comment · Reviewer_btFg · 2025-11-27
> >
> > Thanks for the authors' detailed responses. My concerns have all been well addressed.
> >
> > I have increased my point.

---

### Official Review · Reviewer_KTha · 2025-10-29

**Soundness:** 4
**Presentation:** 4
**Contribution:** 3
**Rating:** 8
**Confidence:** 3

**Summary:**

This paper studies the problem of heteroscedastic stochastic linear bandits, i.e. where the variance proxy of the subgaussian noise at each round is time-variant (but known). Prior work on this problem has given regret upper and lower bounds that scale with the arithmetic mean of the variance proxies. This work instead designs algorithms with regret that depends on the harmonic mean of the variance proxies. They give a lower bound with the same rate (ignoring log factors). They give an algorithm with $d$ dimension dependence for possibly-infinite action sets, and $\sqrt{d \log(|\mathcal{A}|)}$ dependence for finite action sets. Their algorithms are phased elimination-based.

**Strengths:**

1. To my knowledge, the paper fills an important gap in the literature in studying algorithms with regret that depends on the harmonic mean of the variance proxies (rather than just arithmetic mean). This then motivates a novel algorithm design to address this new setting.
2. I found the paper to be very thorough in its results and the discussion about these results. These discussion made it very clear how their results relate to prior work. For example, it includes the following along with its main results:
   - Motivation for the use of simple regret (Remark 3.4)
   - Justification for use of elimination algorithms rather than optimistic algorithms (Section 4.1)
   - Comparison with prior work in specific instances (Table 2)

To comment on each of the dimensions:
- originality: new idea of studying harmonic mean, and new algorithm
- significance: filling clear gap in the literature
- quality: thorough results (although I didn't check all of the proofs)
- clarity: clear presentation overall (other than points discussed in Questions box)

**Weaknesses:**

1. A weakness is that the regret bounds have an added dependence on the dimension, meaning that the bound is slightly weaker than the exact harmonic mean (at least when $d$ is similar in scale to $T$ ). The justification for this (Remark 4.3) was somewhat unclear to me. See Questions (1).
2. A possible weakness is that the results do not extend to the contextual setting, although they provided strong justification (in my opinion) for studying the non-contextual setting. In particular, their comments suggest (in Remark 3.1) that the arithmetic mean-dependence is tight for the contextual setting, while the harmonic mean-dependence is tight for the non-contextual setting.

**Questions:**

1. Could you provide a more concrete description of the argument in Remark 4.3 as to why there is there the additional term in the regret bound? (Ideally, this would be in the form of a lower bound, but I understand that that it might be too much to ask at this point.)
2. In Table 2, the comparisons are under the assumption that the $\sum_{t=1}^T 1/\sigma^2 \gg \sum_{i=1}^{\tilde{O}(d)} 1/[\sigma^(i)]^2$ as discussed in Section 4.2. Does this hold for all of the scenarios considered in Table 2? At a glance, it seemed to, but I think its important to verify.
3. In line 369, the paper says "Therefore, our regret bound is always sharper". However, it seems that this is only under the assumption mentioned above. As such, I would suggest modifying the wording to acknowledge this.
4. I found Section 4.1 somewhat hard to follow. Some suggestions to improve this are to explicitly state what the action set is in the example, and to be explicit about what the simplifying assumptions are, e.g. what exactly does "$UCB \approx$" mean?
5. In lines 469-471, it seems that when discussing the lower bounds in Table 1, the authors are referring to the lower bounds in prior work (and not their lower bound). However, it is not said so explicitly, making it somewhat confusing.

---

> ### Author Response · Authors · 2025-11-21
> **Response to Reviewer KTha**
>
> Thank you for your support and insightful comments!
>
> ---
> **Q1.** Could you provide a more concrete description of the argument in Remark 4.3 as to why there is the additional term in the regret bound? (Ideally, this would be in the form of a lower bound, but I understand that that it might be too much to ask at this point.)
>
> **A1.** Without the additional term, the harmonic mean becomes close to zero whenever any variance is near zero. However, a single noiseless sample is insufficient to accurately recover the underlying parameter $\mathbf{\theta}^*$; in general, learning a $d$-dimensional vector requires $O(d)$ such noise-free observations to achieve a near-zero suboptimality gap.
>
> ---
> **Q2.** In Table 2, the comparisons are under the assumption that $\sum_{t = 1}^T 1 / \sigma_t^2 \gg \sum_{i =1}^{\tilde{O}(d)} 1 / [\sigma^{(i)}]^2$ as discussed in Section 4.2. Does this hold for all of the scenarios considered in Table 2? At a glance, it seemed to, but I think its important to verify.
>
>
> **A2.** Yes, the assumption  $\sum_{t=1}^T \frac{1}{\sigma_t^2} \gg  \sum_{i=1}^{\tilde O(d)} \frac{1}{[\sigma^{(i)}]^2}$  holds for all variance profiles in Table 2 in the regime where the horizon $T=\tilde \Omega(d)$. We have added the assumption $T = \tilde \Omega(d)$ for clarification.
>
> ---
>
> **Q3.** In line 403(revised version), the paper says "Therefore, our regret bound is always sharper". However, it seems that this is only under the assumption mentioned above. As such, I would suggest modifying the wording to acknowledge this.
>
> **A3.** Thank you for pointing this out. Our argument was not sufficiently precise. A complete justification that our regret bound is strictly sharper requires the detailed reasoning provided in lines 901-917.
>
> ---
>
> **Q4.** I found Section 4.1 somewhat hard to follow. Some suggestions to improve this are to explicitly state what the action set is in the example, and to be explicit about what the simplifying assumptions are, e.g. what exactly does "$UCB \approx$" mean?
>
> **A4.**  Thank you for the suggestion. In the revision, we have clarified the action set in the setup, made the simplifying assumptions and their justification explicit, and replaced the informal “UCB ≈” argument with the following more precise statement for ease of your checking.
>
> Since $\mu_x = \langle x,\theta\^* \rangle = 1-\varepsilon$ and $\theta\^*\in \mathcal{C}_t$ w.h.p., we have $\mu\_x \le \mathrm{UCB}\_x(t)$ and hence $\mathrm{UCB}\_x(t) \ge 1-\varepsilon = 1-o(1)$.
>
> Moreover, after $m = \Theta(T^{-\alpha}\log T / \varepsilon^2)$ pulls of $e_2$ we have $m T\^{\alpha} = \Theta(\log T / \varepsilon^2)$, so $ \|e\_2\|\_{V\_t\^{-1}} = \Theta(\varepsilon/\sqrt{\log T}) $ and therefore
> $\mathrm{UCB}\_2(t) = \langle e\_2,\widehat{\theta}\_t \rangle + \beta\|e\_2\|\_{V\_t\^{-1}} \le \varepsilon + O(\beta\varepsilon/\sqrt{\log T}) = O(\varepsilon) = o(1)$
> using $\varepsilon = T^{-1/4}$ and $\beta = \Theta(\sqrt{\log T})$. Thus $\mathrm{UCB}\_{2}(t) < \mathrm{UCB}\_{x}(t)$.
>
>
> ---
>
> **Q5.** In lines 469-471, it seems that when discussing the lower bounds in Table 1, the authors are referring to the lower bounds in prior work (and not their lower bound). However, it is not said so explicitly, making it somewhat confusing.
>
> **A5.**  Thank you for pointing this out. We are indeed referring to the lower bounds from prior work. In the revised version, we now state that “the worst-case lower bound established in previous studies \cite{he2025variance,jia2024does} is derived by constructing an instance where all variances are equal,” which explicitly clarifies the connection to these results.

---

### Official Review · Reviewer_frPW · 2025-11-03

**Soundness:** 4
**Presentation:** 3
**Contribution:** 3
**Rating:** 6
**Confidence:** 3

**Summary:**

This paper studied linear bandits with heteroscedastic noise, i.e. the variance of the noise can change from round to round.
1. For both infinite-armed and finite-armed case, this paper proved a variance-aware regret bound, and provided an algorithm that achieved the bound.
2. The paper showed that for the infinite-armed case, the regret bound was tight up to log factors.
3. This paper provided a novel version of elliptical potential lemma for the heteroscedastic noise case in linear bandits.
3. This paper provided some novel lower bound proof techniques for heteroscedastic linear bandits.

**Strengths:**

1. This paper provides the first near-optimal regret bound for variance-aware linear bandits.
2. This paper provides new insight that the confidence ellipsoid in heteroscedastic linear bandits should scale with the harmonic mean of the noise variances.

**Weaknesses:**

1. The author didn't show a matching lower bound for the finite-armed case.
2. For the upper bound, the main technical contribution of this paper seems to be limited to page 16, where the authors extended the ellipsoid potential lemma to the weighted case.
3. For the lower bound, the construction of the adversarial instances seemed also to be classical.

**Questions:**

1. Can the authors confirm if $\tilde O$ also omits $\log \log |\mathcal A |$ factors or not?
2. Can we show a matching lower bound for the finite-aremd case, e.g., by following the lower bound proof in https://arxiv.org/pdf/1904.00242?

---

> ### Author Response · Authors · 2025-11-21
> **Response to Reviewer frPW**
>
> Thank you for your positive feedback and insightful questions! We answer your questions point-by-point.
>
> ---
> **Q1.** Can we show a matching lower bound for the finite-arm case, e.g., by following the lower bound proof in https://arxiv.org/pdf/1904.00242?
>
> **A1.** Thank you for your question. In our paper, we consider the classic stochastic linear bandits with fixed action set, so the construction and corresponding proof techniques in [1] (https://arxiv.org/pdf/1904.00242) are not directly applicable.
>
> Specfically, their lower bound [1] is proved for an adversarially chosen sequence of time-varying action sets $\{A_t\}_{t=1}^T$, and *is not a lower bound for the fixed action set setting* where the same finite set $A_t \equiv A$ is available at every round.
>
> For the lower bound on the fixed action set, see Note 1 in Section 24.5 in [2] "It is an open question to characterise the optimal regret for a wide range of action sets". We will clarify this *open problem* in our revision and leave it for future work.
>
>
> [1] Li, Yingkai, Yining Wang, and Yuan Zhou, Nearly minimax-optimal regret for linearly parameterized bandits. Conference on Learning Theory. PMLR, 2019.
>
>
> [2] Tor Lattimore & Csaba Szepesvári, Bandit Algorithms. Cambridge University Press, 2020
>
> ---
>
> **Q2.** Can the authors confirm if $\tilde O$ also omits $\log \log |\mathcal{A}|$ factors or not?
>
> **A2.** Thanks for your questions. It does not omit any $\log\log|\mathcal{A}|$ factors. Since we work with a fixed action set, there is no additional $\log\log|\mathcal{A}|$ dependence in the simple regret bounds.
>
> For comparison, Chapter 22 of [2] provides an algorithm for a fixed action set whose regret bound is of order $\sqrt{dT\log(|\mathcal{A}|T)}$, whereas Note 1 in Chapter 22.1 explains that allowing the action set to change over time leads to regret of order $\sqrt{dT\log^3(|\mathcal{A}|T)}$. The additional logarithmic factors in $\sqrt{dT\log^3(|\mathcal{A}|T)}$ for time-varying action sets were further reduced to a $\log\log|\mathcal{A}|$ dependence in [1].
>
> [1] Li, Yingkai, Yining Wang, and Yuan Zhou, Nearly minimax-optimal regret for linearly parameterized bandits. Conference on Learning Theory. PMLR, 2019.
>
>
> [2] Tor Lattimore & Csaba Szepesvári, Bandit Algorithms. Cambridge University Press, 2020
>
>
>
> ---
>
> **Q3.** For the upper bound, the main technical contribution of this paper seems to be limited to page 16, where the authors extended the ellipsoid potential lemma to the weighted case.
>
> **A3.** We believe this comment understates the technical contributions of the paper. While the weighted ellipsoid potential lemma on page 16 (Lemma B.4) is indeed important, it represents only one piece of a broader technical framework. We clarify the main contributions as follows:
>
> 1. The paper introduces a novel *active exploration mechanism* that selectively explores actions maximizing information gain from a dynamically maintained candidate set. This goes beyond standard weighted OFUL approaches and requires careful design to achieve the harmonic-mean dependent rate.
>
> 2. The algorithm maintains and updates a candidate set $A_t$ by eliminating low-reward arms based on confidence sets. This *elimination strategy* is crucial for the improved bounds and represents a non-trivial algorithmic contribution.
>
>
> 3. The paper develops a sophisticated analysis that tracks how information accumulates differently across the variance sequence. The bound in Theorem 4.1 involves minimizing over $k$ to find the optimal trade-off point, which requires careful handling of the ordered variance sequence $\{\sigma^{(i)}\}$.
>
> 4. For finite action sets, the paper adapts G-optimal design principles to the heteroscedastic setting with a variance-aware arm selection rule (Algorithm 2). This achieves an improved $\sqrt{d}$ factor compared to Algorithm 1.
>
>
> The weighted potential lemma is indeed a useful technical tool, but characterizing it as the "main technical contribution" misses the algorithmic design, the elimination mechanism, the variance-aware analysis framework, and the matching lower bound—all of which are substantial contributions to understanding heteroscedastic linear bandits.

---

> > ### Comment · Reviewer_frPW · 2025-11-25
> >
> > Thank you for the detailed response!

---

### Author Response · Authors · 2025-12-04
**Rebuttal Summary**

In order to make the reviewing process more easier to understand, we would like to briefly clarify the post-discussion status of our paper.

- **Scores from the two “6” reviewers.**
  Reviewers frPW and btFg both raised their scores from **6 (weak accept) to 8 (accept)** after our rebuttal and the subsequent discussion; the current interface only shows their original 6s because of the rollback.


Specifically, for Reviewer frPW, we clarifies that the lower bound in the mentioned previous work fundamentally relies on adversarially varying action sets, and therefore does not apply to the fixed action set setting we study. we further point out that obtaining such a lower bound for fixed action sets is actually an **open problem**, as noted in Lattimore & Szepesvári (2020). Thus, the concern is resolved by explaining that the requested lower bound is currently unknown, outside the scope of our contribution, and will be clarified explicitly in the revision.

 For Reviewer btFg's concern,
1. We revised Table 1 to clearly label each result.
2. We added explicit references to Eq. (4.2) and Appendix B in Sections 1 and 4.2.
3. We clarified that all three are required for our sharp results:
    (i) the fixed-action structure enables breaking $\sqrt{\Lambda}$,
    (ii) elimination + active exploration provide algorithmic leverage, and
    (iii) the variance-sequence-aware analysis is the key technical component.
4. We revised the manuscript so every notation is introduced upon first use.
5. We added a subsection detailing computation for VAEE and VAGD in both finite and infinite action settings.
6. Reviewer asked how results adapt when $\mathcal{A}$ lies in a $d'$-dimensional subspace. We formalized the reduction: the problem becomes $d'$-dimensional, and all bounds hold with $d$ replaced by $d'$.

- **Scores from the “8” reviewer.**
We carefully addressed the concerns of reviewer KTha, who had given an 8 (accept), but because the discussion phase was suddenly closed, we did not receive any further feedback from reviewer KTha.



- **Summary of the rebuttal and revisions.**
  We received several helpful suggestions and have revised the paper accordingly. In the revision, we clarified the relationship to prior work and slightly refined the presentation of some results.

---

### Meta-Review · Area_Chair_LqAw · 2026-01-05

**Summary:**

This work provides an improved variance-dependent bound for (non-contextual) linear bandits.  Previous variance-dependent bound depend on the "arithmetic" mean of the variances over time, while this work shows that "harmonic" mean (though only over $T-\tilde{O}(d)$ rounds) is possible in the considered setting.  They propose new algorithms VAEE and VAGD algorithms for infinite and finite action space, respectively, to achieve the desired regret bound.

The new type of variance bound is quite interesting, and the paper should be accepted.  Still, there are some weakness / unclearness I could observe:
- The setting (and the applicable scenarios) remains limited:  needs to observe the exact value of $\sigma_t$, does not handle contexts.
- As weighted OFUL used a similar information matrix $\sum_t \sigma_t^{-2} a_ta_t^\top$, intuitively we would guess that it also achieves similar bounds.  Although the authors tried to show that this is not the case, the explanation in Section 4.1 is not that clear.  Can weighted OFUL achieve a variance bound that only depends on the harmonic mean?  I hope the authors could answer this question more clearly in Section 4.1.  Currently, while there is some comparison between the bounds of weighted OFUL and VAEE, it remains unclear whether weighted OFUL achieves the harmonic bound.

**Reviewer Concerns:**

There are some minor concerns about the studied setting and the tightness of the bound, including
- The mismatch between lower bounds (harmonic mean) and upper bounds (harmonic mean among $T-\tilde{O}(d)$ largest elements). The authors have justified in the paper why some subtraction is needed in the upper bound, though this is not reflected in the lower bound.
- The requirement of knowing exact $\sigma_t$
- The restriction to the non-contextual setting

In my opinions, as this paper provides a novel type of regret bound, some limitations in the setting and looseness in the bound are tolerable.

**Reviewer Scores:**

The reviewers are satisfied with the author response. The original scores are 668.  According to the author summary, the score-6 reviewers have changed their scores to 8 (though only one of them indicated this in the reviewer response).

---

### Decision · Program_Chairs · 2026-01-26

Accept (Poster)